# Symbolic regression via MDLformer-guided search: from minimizing prediction error to minimizing description length

**Zihan Yu**
Department of Electronic Engineering, BNRist
Tsinghua University
Beijing, China

**Jingtao Ding**[*]
Department of Electronic Engineering, BNRist
Tsinghua University
Beijing, China

**Yong Li**[*]
Department of Electronic Engineering, BNRist
Tsinghua University
Beijing, China

**Depeng Jin**
Department of Electronic Engineering, BNRist
Tsinghua University
Beijing, China

## Abstract

Symbolic regression, a task discovering the formula best fitting the given data, is typically based on the heuristical search. These methods usually update candidate formulas to obtain new ones with lower prediction errors iteratively. However, since formulas with similar function shapes may have completely different symbolic forms, the prediction error does not decrease monotonously as the search approaches the target formula, causing the low recovery rate of existing methods. To solve this problem, we propose a novel search objective based on the minimum description length, which reflects the distance from the target and decreases monotonically as the search approaches the correct form of the target formula. To estimate the minimum description length of any input data, we design a neural network, MDLformer, which enables robust and scalable estimation through large-scale training. With the MDLformer's output as the search objective, we implement a symbolic regression method, SR4MDL, that can effectively recover the correct mathematical form of the formula. Extensive experiments illustrate its excellent performance in recovering formulas from data. Our method successfully recovers around 50 formulas across two benchmark datasets comprising 133 problems, outperforming state-of-the-art methods by 43.92%. Experiments on 122 unseen black-box problems further demonstrate its generalization performance. We release our code at `https://github.com/tsinghua-fib-lab/SR4MDL`.

## 1 Introduction

Symbolic regression (SR) is a task that uncovers interpretable mathematical formulas to describe the underlying relationships within observational data, which is widely used for promoting scientific discovery or facilitating the modeling of diverse phenomena in many fields, such as dynamical systems (Quade et al., 2016; Chen et al., 2019; Cornelio et al., 2023; Angelis et al., 2023; Ding et al., 2024), materials science (Wang et al., 2019; Schmelzer et al., 2020; Weng et al., 2020; Sun et al., 2019), etc. (Liu et al., 2024; Neumann et al., 2020; Shi et al., 2022). Formally, SR aims at finding a symbolic function $f$ from the given data $(x, y)$, where $x = [x_1, x_2, ..., x_D] \in \mathbb{R}^{N \times D}$ and $y \in \mathbb{R}^{N \times 1}$ are observed $N$ samples points of independent and dependent variables. The discovered formulas consist of mathematical symbols like $+, -, \times$, and $\div$, whose specific form can provide corresponding insights into the patterns behind the data. To find the formula that best fits the data in the symbolic space, the most typical methods to SR are based on heuristic search, in particular, the genetic programming (GP) algorithm (Augusto & Barbosa, 2000), which executes evolution

---

[*]Corresponding author. Email: `dingjt15@tsinghua.org.cn`, `liyong07@tsinghua.edu.cn`

iteratively to enhance the fit of candidate formulas to the given data. There is now a large body of commercial and open-source software (Dubčáková, 2011; Stephens, 2016; Cranmer, 2023), as well as many influential works (Schmelzer et al., 2020; Weng et al., 2020; Liu et al., 2024), that developed based on the heuristic search-based SR methods.

However, while existing SR methods can identify formulas with high accuracy ($R^2 > 0.99$ for over $90\%$ of cases in SRbench (Cavalab, 2022)), their effectiveness in discovering the optimal formula with the lowest prediction error is limited (only around $20\%$ success rate), even in noise-free data. This is because these methods optimize the candidate formulas' prediction errors (Makke & Chawla, 2024), which does not lead to the target formula with the minimum prediction error. As illustrated in Figure 1, the mean squared error of a candidate formula ($f_i$) does not decrease monotonically when its form approaches the target one. This is because formulas with similar symbolic structures can exhibit rather different functional shapes in the numerical space. On the other hand, two formulas with similar functional shapes can have entirely different symbolic forms. This indicates that the SR task lacks an *optimal substructure* (Cormen et al., 2022), that is, the target formula with the minimum prediction error cannot be achieved by simply making small adjustments to formulas with sub-optimal prediction errors. This high nonlinearity of the relationship between a formula's mathematical form and its functional shape results in a divergence between the direction of reducing prediction error and the direction toward the target formula. This complexity makes it difficult for existing methods to identify the formula with correct mathematical forms.

To solve this problem, in this work we proposed a new search objective inspired by the minimum description length (MDL) (Kolmogorov, 1963), which represents the size of the simplest model used to describe the data, or, in SR, the minimum number of symbols required for the target formula $y = f(x)$. If each mathematical symbol is regarded as a transformation step, then MDL describes the number of transformations needed to go from the independent variable, $x$, to the dependent variable, $y$. Therefore, the search with minimal MDL as the optimization objective has an optimal substructure (Cormen et al., 2022): only when the correct transformation is executed can the MDL reduce, thus the search direction of MDL reduction is always consistent with the direction leading to the target formula (see Figure 1). However, MDL has been shown to be incomputable (Vitányi, 2020), indicating that no algorithm can provide an accurate MDL for arbitrary input. Nevertheless, the capacity of neural networks as universal approximators (Nielsen, 2015) indicates that, with sufficient data, a suitably designed neural network can effectively learn the complex mapping from the data to its MDL. To this end, we developed a Transformer-based neural network, MDLformer. Through large-scale training on over 130 million symbolic-numeric pairs using a carefully designed training strategy that aligns the numerical space with the symbolic space, MDLformer has gained the ability to estimate the MDL of any given data. With the search objective provided by the large-scale trained MDLformer, we implement a new SR method based on the Monte Carlo tree search algorithm (Browne et al., 2012), a heuristic search algorithm that has been proven to be suitable for use in conjunction with neural networks successfully by many works (Silver et al., 2016; 2017; Kamienny et al., 2023).

We conduct extensive experiments to illustrate the excellent performance of our approach in recovering formulas from data. Across two problem sets with 133 formulas, our method successfully recovers around 50 of them, outperforming state-of-the-art methods by 43.92%. We also find this result robust to noise: even if we add noise with an intensity of $10\%$ to the data, the recovery rate of our method is still higher than the recovery of other methods in the absence of noise. We also test our method on the black-box problem sets, finding it can discover formulas that describe the data with lower description length and higher accuracy than other methods. Further analysis of the MDLformer demonstrates its scalable and robust capability for accurate predictions of the MDL, which explains the outstanding performance of our SR method.

## 2 RELATED WORK

**Heuristic search methods.** Traditionally, symbolic regression approaches are mainly based on the genetic programming (GP) algorithms(Langdon & Poli, 2013), which maintains a set of symbolic formulas and iteratively updates these candidate formulas via mutation and crossover operations. To this day, there have been plenty of GP-based symbolic regression toolkits developed, such as Eureqa (Dubčáková, 2011), GPlearn (Stephens, 2016), PySR (Cranmer, 2023), and so on (Schmidt & Lipson, 2010; de Franca & Aldeia, 2021; La Cava et al., 2016; Arnaldo et al., 2014; Virgolin et al., 2019; 2021; Burlacu et al., 2020; Zhong et al., 2018; Zhang et al., 2022; Augusto & Barbosa, 2000; Kartelj

Figure 1: **Comparison of the two search objectives.** In the searching route leading to the target, $f^*$, the prediction error (measured by the mean square error, MSE) does not decrease monotonically as the candidate formula's form gets closer to the target one, whereas the minimum description length (MDL) does. Here, $\phi_i$ denotes the function $f^* = \phi_i(x, f_i)$ and $C[\phi_i]$ is its complexity.

& Djukanović, 2023; Smits & Kotanchek, 2005; Searson et al., 2010). In these years, people have begun to use reinforcement learning algorithms for symbolic regression, including Monte-Carlo tree search (MCTS) (Sun et al., 2022), double Q-learning (Xu et al., 2024), and deep reinforcement learning (DRL) (Petersen et al., 2021; Tenachi et al., 2023). These methods start from the empty formula and iteratively select appropriate mathematical symbols to fill in until a complete formula is obtained. Some works combine multiple methods to achieve better symbol regression (Mundhenk et al., 2021; Jin et al., 2020; McConaghy, 2011; Landajuela et al., 2022). These search algorithms, however, all face the problem that the prediction errors usually do not decrease monotonically from the initial state to the ground-truth formula. Although recent methods introduced formula complexity as a regularization term alongside the prediction error (Sun et al., 2022; Cranmer, 2023), this problem persists and their recovery rates remain low. In contrast, our approach addresses this problem by optimizing the minimum description length (MDL), which decreases monotonically along the path to the target formula.

**Neural network-assisted search methods.** Recently, some works have attempted to enhance the search methods by utilizing the powerful fitting capabilities of neural networks. For example, Mundhenk et al. (2021) proposes to train an RNN by deep reinforcement learning to generate the initial candidate formulas in genetic algorithms for speeding up evolution. While Kamienny et al. (2023) utilizes a Transformer pre-trained as a next-symbol predictor to provide promising search directions for the Monte Carlo tree search algorithm. Although these works still take prediction error as the search goal and thus do not have optimal substructure, they inspire us to combine pre-trained neural networks with search algorithms. AIFeynman shows another way to combine neural networks and search algorithms (Udrescu & Tegmark, 2020; Udrescu et al., 2020): it discovers symmetry properties in data by fitting it with neural networks and, based on this, converts the search for the target formula into searches for several smaller subformulas. However, leveraging hand-designed symmetry properties summarized on the Feynman dataset, AIFeynman cannot be adapted to other datasets without these properties, and these rules are sensitive to noise, making AIFeynman's recovery rate decrease significantly on noisy data (Cavalab, 2022). As a comparison, our method can also be regarded as a method of recursively simplifying the target function, since the decrease of MDL indicates a candidate formula $f$ as a subformula of the target one. Identifying subformulas based on MDLformer's output rather than hand-designed rules, our method does not rely on specific prior knowledge and is more robust to noise.

**Regression and generative methods using neural networks.** In addition to using neural networks to assist search, some works proposed neural network-based regression methods. They design fully connected neural networks with mathematical functions, such as $\sin, \exp, \times$, as activation layers. After fitting weight parameters to the input data, they can extract mathematical formulas from the networks Kim et al. (2020); La Cava et al. (2019); Kubalík et al. (2023). However, due to using mathematical functions like exponential and logarithmic functions as activation layers, these methods usually face numerical instability problems such as gradient explosion. Other works are based on generative methods. They use large-scale pre-trained Transformers to generate symbolic sequences as target formulas from the data directly.(Biggio et al., 2021; Kamienny et al., 2022; Meidani et al., 2023; Bendinelli et al., 2023; d'Ascoli et al., 2022). Once pre-trained, these methods can directly generate target formulas without searching or extra training. Therefore, they are usually faster than other methods. However, it is still difficult for them to obtain the target formula with the correct form, since small changes in the input data can lead to completely different objective functions (Kamienny et al., 2023). As a comparison, based on the search method, our approach can improve the results through extended running durations, and the utilization of the MDL search objective enhances its efficiency, allowing it to achieve a high recovery rate for the target formula

within a reasonable timeframe. Many recent studies focus on neural symbolic reasoning, i.e., using neural networks for symbolic reasoning(van Krieken et al., 2023). Despite their similar names, this task is different from symbolic regression, which we detailed discussed in Appendix A.

# 3 LEARNING TO ESTIMATE FORMULA COMPLEXITY WITH MDLFORMER

## 3.1 MDLFORMER

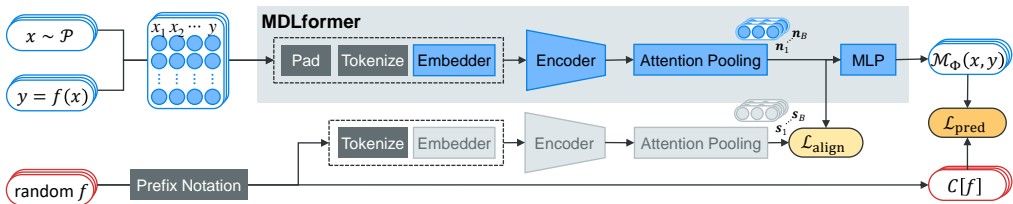

Figure 2: **Schematic diagram of the architecture and learning process of MDLformer.**

The design of MDLformer is rooted in the recent development of Transformer models, which have been proven to be able to encode numeric observation data into latent space, which can be further used for inferring corresponding symbolic formulas (Kamienny et al., 2022). Meidani et al. (2023) further find that, by aligning the latent spaces of two Transformer models that encode numeric data and symbolic formulas, respectively, it's possible to predict cross-model properties. Here we exploit this approach to cross-modally predict the length of the corresponding symbolic formula, i.e., the minimum description length (MDL), based on numeric observational data,

As depicted in Figure 2, the MDLformer, $\mathcal{M}_\Phi$, integrates an embedder, a Transformer encoder, an attention-based pooling, and a multilayer perceptron (MLP) as readout head, to map the input data $x \in \mathbb{R}^{N \times D}$ and $y \in \mathbb{R}^{N \times 1}$ into its MDL, $C[f]$, where $f$ denotes the simplest symbolic function that describes $y = f(x)$ and $C[f]$ is its complexity, i.e., the number of symbols required to express $f$.

**Embedding.** We first pad $x$ into $\mathbb{R}^{N \times D_{\max}}$ with zeros since the number of features $D$ can vary. We then tokenize the input data using base-10 floating-point notation. Specifically, we round each value to 4 significant digits and then split it into three parts: sign (+, -), mantissa ($0.000 \sim 9.999$), and exponent ($E$-100 $\sim$ $E$+100). For example, a value $54.321$ is represented as a sequence [+,5.432,E+1]. The input data is tokenized into $N \times (D_{\max} + 1) \times 3$ tokens, which, increases with $N$ and $D_{\max}$, challenges the quadratic complexity of Transformers. Therefore, we sample no more than $N_{\max}$ pairs from each row of $x$ and $y$ and then embed each x-y pair into the latent space with a fixed dimensionality $d_f$ to feed into the Transformer encoder.

**Encoding.** We leverage a Transformer encoder (Vaswani et al., 2017) to process the embedded numeric input. Notably, we remove the positional encoding in the numeric encoder since each item in the input sequence represents a pair of x-y data and their order is thus not important, which aligns with the previous practices (Biggio et al., 2021; Kamienny et al., 2022; Meidani et al., 2023).

**Pooling.** To map the Transformer encoder's outputs, $V \in \mathbb{R}^{N \times d_f}$, into a fixed-size representation, we adopt an attention-based pooling mechanism (Santos et al., 2016). Specifically, we use a learnable weight, $w \in \mathbb{R}^{d_f}$, to calculate an attention weight for each row $v_i \in V$ by $a_i = \mathrm{softmax}(w \cdot v_i)$, where the softmax is conducted along the sequence dimension $N$. Then, we add them up with $a_i$ as weights to get the final representation: $n = \sum_i a_i v_i$.

**Reading out.** After pooling we obtain a compact representation for the whole numeric input, which can be used to predict cross-model properties (Meidani et al., 2023). Here we predict the MDL with a multilayer perceptron that uses ReLU as the activation layer.

## 3.2 TRAINING DATA GENERATION

The training of the MDLformer relies on plenty of paired numeric and symbolic data, which is generated during the learning process on the fly. During the whole process, we generate a total of about 131 million pairs of input data.

**Generate symbolic formulas.** We generate symbolic formulas with the algorithm proposed by Kamienny et al. (2022) to ensure diversity: First, we sample the number of features, $D \sim$

$\mathcal{U}\{0, \cdots, D_{\max}\}$. Then, we randomly combine them with $b$ binary operators and further sample $u$ unary operators to insert into the random position of the resulting formula. Finally, we add non-similar transformations to each position of the formula: $f_i \mapsto a_i f_i + b_i$, where $f_i$ are subformulas of $f$, $a_i$ and $b_i$ are sampled from $\mathcal{U}[-100, 100]$. To ensure that $f$ is in its simplest form, we simplify it with sympy, an open-source Python package for algebra processing. The number of operators, variables, and parameters in the simplified $f$ is its length, and therefore, is the MDL of $(x, y = f(x))$.

**Generate numeric data.** After generating the formula, we sample the number of features, $D$, from $\mathcal{U}\{1 \cdots D_{\max}\}$ and generate numeric data for $D$ independent variables: $x \in \mathbb{R}^{N \times D} \sim \mathcal{P}$. Then, we calculate corresponding numeric data for the dependent variable: $y = f(x) \in \mathbb{R}^{N \times 1}$. Considering that $f$'s domain cannot be the whole $\mathbb{R}^D$ when it contains specific operators, such as logarithm or square root, some of the sample points in $x$ may not be in its domain and lead to invalid dependent variable values. Therefore, we discard invalid values in y and the corresponding rows in x, ensuring each sample point of $x$ lies in the domain of $f$. We also find that the choice of $\mathcal{P}$ can significantly influence the performance of MDLformer when using it for symbolic regression. We use the Gaussian mixture model (GMM) as suggested by Kamienny et al. (2022), which takes the sum of $C$ Gaussian distributions with random mean and variance and normalizes them as the distribution $\mathcal{P}$. To further improve the diversity of numerical data and enhance its prediction performance in actual symbolic regression, we also considered another sampling method, that is, generating it by a function transformation on hidden variables $z$. Specifically, we first sample $z \in \mathbb{R}^{N \times K}$ from a GMM distribution, and then generate a random symbolic function $g \sim \mathcal{F}_{\mathcal{K} \times \mathcal{D}}$ following the methods introduced above, and finally operate the hidden variable with the function to obtain $x = g(z) \in \mathbb{R}^{N \times D}$. This design comes from the fact that, when using MDLformer for symbolic regression, we need to estimate the minimal description length of the transformation results obtained by operating symbolic functions on the input data $x$.

## 3.3 Cross-Modal Learning for Formula Complexity

With the generated formulas $f_i$ and data $(x_i, y_i)$, we train the MDLformer using two learning objectives, including a primary objective that predicts the accurate MDL and an auxiliary objective that aligns the numeric latent space and the symbolic latent space, as depicted in Figure 2.

**Primary learning objective for prediction.** To train the MDLformer for predicting the corresponding minimum description length (MDL) based on numerical input, we optimize the mean square error, $\mathcal{L}_{\text{pred}}$, between MDLs estimated by the MDLformer and ground-truths:

$$\mathcal{L}_{\text{pred}} = \frac{1}{B} \sum_{i=1}^{B} \left( C[f_i] - \mathcal{M}_\Phi(x_i, y_i) \right)^2, \tag{1}$$

where $B$ denotes the batch size, $C[f_i]$ is the complexity of the symbolic function $f_i$.

**Alignment as an auxiliary learning objective.** The auxiliary objective, proposed by Meidani et al. (2023), aims to facilitate a mutual understanding of both numeric and symbolic domains and thus empower better cross-modal prediction. Specifically, as suggested by Meidani et al. (2023), we introduce a symbolic encoder to map the prefix notation of $f$ into a compact representation $s$, which has a structure similar to the numeric encoder, as depicted in Figure 2. The latent spaces of these two encoders are aligned by optimizing a symmetric cross-entropy loss over similarity scores:

$$\mathcal{L}_{\text{align}} = -\left( \sum_{i=1}^{B} \log \frac{\exp(\boldsymbol{n}_i \cdot \boldsymbol{s}_i / \tau)}{\sum_{j=1}^{B} \exp(\boldsymbol{n}_i \cdot \boldsymbol{s}_j / \tau)} + \sum_{i=1}^{B} \log \frac{\exp(\boldsymbol{s}_i \cdot \boldsymbol{n}_i / \tau)}{\sum_{j=1}^{B} \exp(\boldsymbol{s}_i \cdot \boldsymbol{n}_j / \tau)} \right), \tag{2}$$

where $B$ is the batch size, $\tau$ is the temperature parameter, $\boldsymbol{s}_i$ and $\boldsymbol{n}_i$ are encoded representations of $i$-th numeric data and symbolic function, respectively. Note that this loss is calculated per batch.

**Training Strategy** We adopt a two-step training strategy: First, we train the MDLformer with the auxiliary objective and the generated formula-data pairs, where the numeric data is sampled from the GMM. ii) Then, we use the primary objective to train the MDLformer on the numeric data generated by operating symbolic functions on latent variables. The first step aligns the latent space of the numerical encoder and the symbolic encoder and thus provides a good start for the numerical encoder to predict the formula length in the symbolic space cross-modally. Based on this, the second step uses a data distribution that is closer to the actual symbolic regression used, allowing the data encoder to estimate the description complexity of the input data accurately.

# 4 MDL-GUIDED SYMBOLIC REGRESSION WITH MDLFORMER

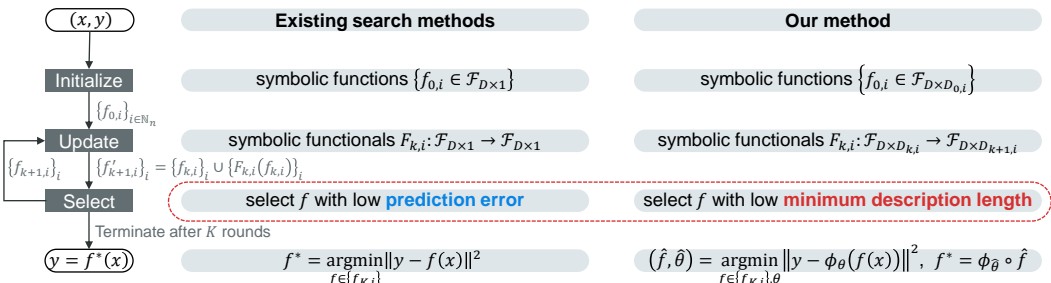

Figure 3: **Comparison of existing search methods and our method.** $x \in \mathbb{R}^{N \times D}$ and $y \in \mathbb{R}^{N \times 1}$ are observational data, $\mathcal{F}_{p \times q}$ denotes all possible symbolic functions mapping from $\mathbb{R}^{N \times p}$ to $\mathbb{R}^{N \times q}$. $i \in \mathbb{N}_n$ denotes the indexes of candidate formulas (in a total of $n$), while $k$ denotes the number of loops (terminated at round $K$). $T_{k,i}$ are algorithm-specific symbolic functionals, like crossover and mutation operations in GP, whose concrete operations in different search algorithms are summarized in Table 3. $\phi_\theta$ are simple functions (like linear functions) with parameters $\theta$ that map $f$ into $\mathcal{F}_{D \times 1}$.

In this section, we explain how the trained MDLformer can help symbolic regression discover target formulas with correct mathematical forms more easily. The main idea is to change the search direction from the direction of prediction error reduction to the direction of minimum description length (MDL) reduction. This leads to the property of optimal substructure and makes it easier for the existing search algorithms to search for the target formula with the correct form.

## 4.1 SEARCHING FOR MINIMUM DESCRIPTION LENGTH

As depicted in Figure 3, existing search-based methods focus on the update-and-select loops over a set of candidate formulas: after generating an initial formula set, $\{f_{0,i} \in \mathcal{F}_{D \times 1}\}$, these methods generate algorithm-specific symbolic functionals $F_{k,i} : \mathcal{F}_{D \times 1} \to \mathcal{F}_{D \times 1}$ (such as the crossover and mutation in genetic programming, see D for details.) to creating new candidate formulas: $f'_{k+1,i} = F_{k,i}(f_{k,i})$, and select resulting functions with low prediction errors for the next round of iteration. This update-and-select loop is repeated iteratively until a formula with a low enough prediction error is encountered, at that point the loop will terminate and this formula will be read out as a result: $f^* = \arg\min_f \|y - f(x)\|^2$.

In contrast, instead of maintaining a candidate set of target formulas, we maintain a candidate set of *subformulas* of the target formula, $\{f_{k,i} \in \mathcal{F}_{D \times D_{k,i}}\}$, where $k$ represents the number of iterations and $i$ represents the sample index. For a candidate $f$, each of its item $f^{(d)}$ denotes a part of the target, i.e., $f^* = \phi(f) = \phi(f^{(1)}, f^{(2)}, \cdots, f^{(D_{k,i})})$. $\phi$ reflects the transformation required from $f$ to the target formula $f^*$, while its complexity, $C[\phi]$, evaluates the "distance" between $f$ and $f^*$. Therefore, by selecting $f$ with low MDL estimated by the MDLformer, $\mathcal{M}_\Phi(f(x), y) \approx C[\phi]$, during the update-and-select loops, the remaining transformation $\phi$ can be simplified iteratively, and thus the candidates $f$ will get "closer" to the target one. For an $f_{K,i}$ that is sufficiently close to $f^*$ after $K$ iterations, the remaining transformation from $f_{K,i}$ to $f^*$ can be described by a simple parameterized function $\phi_\theta$ accurately ($\theta$ are parameters). Formally speaking, we find

$$(\hat{f}, \hat{\theta}) = \arg\min_{f \in \{f_{K,i}\}, \theta} \|y - \phi_\theta(f(x))\|^2 \tag{3}$$

from the candidate set to determine the target formula: $f^* = \phi_{\hat{\theta}}(\hat{f})$.

## 4.2 IMPLEMENTATION

In this work, we implement our method based on the Monte Carlo tree search (MCTS) (Browne et al., 2012) because of its application in symbolic regression task (Sun et al., 2022) as well as its successful combination with neural networks in numerous studies (Kamienny et al., 2023; Silver et al., 2016; 2017). MCTS is a classical heuristic search algorithm that maintains a search tree

with each node representing a candidate formula and updates the tree through four steps: selection, expansion, simulation, and backpropagation. The vast majority of our implementations are consistent with the existing practice of using MCTS for symbolic regression (Sun et al., 2022), except that the upper confidence bound (UCB) to guide the search is based on MDLformer's output (see Appendix D for details). For the remaining transformation $\phi_\theta$, we use the simplest linear function:

$$\phi_\theta(f_{k,i}) = \sum_{d=1}^{D_{k,i}} \theta_d f^{(d)}, \tag{4}$$

where $f^{(d)} \in \mathcal{F}_{D \times 1}$ is the $d$-th item of $f_{k,i}$. Though it is possible to use functions in other forms, such as multiplications or polynomials, or even brute force searches used by AIFeynman (Udrescu & Tegmark, 2020), we find that even the simplest linear functions work well enough as long as MDLformer's estimation is accurate enough.

## 5 EXPERIMENTS

This section is divided into three parts. The first two parts evaluate the effectiveness of our symbolic regression method on 133 ground-truth problems and 122 black-box problems with comparison to state-of-the-art symbolic regression methods. The final part gives an in-depth discussion of the MDLformer's performance in estimating MDL.

### 5.1 SYMBOLIC REGRESSION ON GROUND-TRUTH PROBLEMS

Table 1: **Recovery rate and search time of different methods in both Strogatz and Feynman datasets.** Each experiment is conducted at ten random seeds and four noise levels.

| Type | Method | Strogatz (14 problems) | | Feynman (119 problems) | |
|------|--------|------------------------|------|------------------------|------|
| | | R. Rate ↑ | Time (s) | R. Rate ↑ | Time (s) |
| Regression | FEAT (La Cava et al., 2019) | 0.19% | 636.6 | 0.00% | 1532 |
| Generative | NeurSR (Biggio et al., 2021) | 1.79% | 15.71 | 2.44% | 24.78 |
| | E2ESR (Kamienny et al., 2022) | 3.78% | 4.044 | 10.40% | 4.576 |
| | SNIP (Meidani et al., 2023) | 6.79% | 1.457 | 1.60% | 2.196 |
| Search-based | GPlearn (Stephens, 2016) | 9.21% | 966.2 | 16.89% | 3349 |
| | AFP (Schmidt & Lipson, 2010) | 10.90% | 160.7 | 17.51% | 3845 |
| | AFP-FE (Schmidt & Lipson, 2010) | 12.86% | 9532 | 20.80% | 25138 |
| | EPLEX (La Cava et al., 2016) | 6.02% | 446.8 | 10.10% | 11548 |
| | SBP-GP (Virgolin et al., 2019) | 2.44% | 20089 | 2.88% | 28933 |
| | GP-GOMEA (Virgolin et al., 2021) | 8.46% | 1100 | 10.32% | 3456 |
| | Operon (Burlacu et al., 2020) | 4.29% | 83.58 | 7.97% | 2656 |
| | SPL (Sun et al., 2022) | 8.12% | 363.7 | 10.48% | 263.3 |
| | DSR (Petersen et al., 2021) | 18.05% | 784.3 | 18.60% | 1042 |
| | RSRM (Xu et al., 2024) | 4.43% | 133.2 | 15.40% | 127.1 |
| | AIFeynman2 (Udrescu et al., 2020) | 15.27% | 241.3 | 27.24% | 708.6 |
| | BSR (Jin et al., 2020) | 0.38% | 25346 | 0.70% | 30635 |
| | **Ours** | **66.78%** (+6.82 formulas) | 338.3 | **33.93%** (+7.96 formulas) | 660.5 |

**Experimental Settings** To evaluate the capability of our method to recover formulas from data, we conduct experiments on SRbench (La Cava et al., 2021) as most of the previous work does, which contains 14 Strogatz problems (Strogatz, 2018) and 119 Feynman problems (Udrescu & Tegmark, 2020) with known ground-truth underlying formulas. We compare 16 baseline algorithms split into three types: the generative methods and the regression methods, both of which are based on neural networks, as well as the search methods using genetic programming or reinforcement learning. (see Appendix E.1.1 for details). For each algorithm on each problem, we test its performance from 10 fixed random seeds at 4 different noise levels: $\epsilon = 0.0, 0.001, 0.01, 0.1$.

**Experimental Results** The experiment results are provided in Table 1, where our method reaches the top recovery rate in both problem sets with reasonable search time. In the Strogatz problem

set, our method improves the Recovery Rate from $15.27\%$ to $66.17\%$, increasing nearly three times. In the Feynman problem set, our method also improves the Recovery Rate by $11.94\%$. Overall, our method recovers around 50 formulas out of 133, outperforming the best baseline (around 35 formulas by AIFeynman2) by $43.92\%$. We also observe that search-based algorithms outperform generative and regression methods. This could be because the latter is designed to produce formulas fitting data accurately, rather than formulas with correct forms. On the other hand, the former can improve search results by simply running for a longer period, while the latter cannot(Kamienny et al., 2023), causing the former's higher recovery rates.

In Figure 4 we also plot the recovery rate of all methods at different noise levels (see Appendix E.1.2 for details). On the Strogatz problem set, our method has the highest recovery rate, significantly surpassing other methods. On the Feynman problem set, however, AIFeynman2 shows a higher recovery rate than our methods when the noise level is $0.0$. This is because AIFeynman2 designs a series of rules based on the characteristics of formulas in the Feynman problem set to help search for formulas. In contrast, as a general method with no reliance on any prior knowledge, our method can be directly applied to different problem sets. Furthermore, unlike AIFeynman, which experiences a sharp decline in performance due to the failure of hand-designed rules in the presence of noise, our method is robust to noise: Even at the maximum noise level ($\epsilon = 0.1$), our method still outperforms other methods without noise, including the AIFeynman2 on the Strogatz dataset.

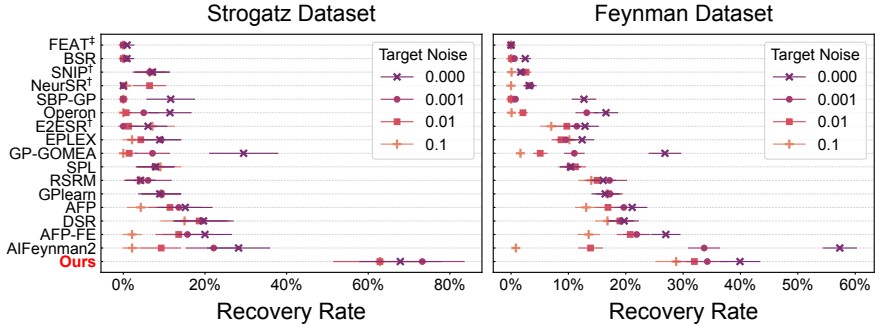

Figure 4: **Recovery rate at different noise levels.** † and ‡ denote generative and regression methods, the others are search methods. The error bars depict the $95\%$ confidence interval.

## 5.2 Symbolic Regression on Black-Box Problems

**Experimental Settings** To verify the generalization ability of our method, that is, whether it can be adopted on data collected from the real world, rather than generated from formulas, we also test our method on 122 black-box problems from Penn Machine Learning Benchmark (PMLB) (Olson et al., 2017). We split the data of each problem into $75\%$ training set and $25\%$ test set, and, instead of recovery rate, we evaluate the results using the Pareto front that balances the test set accuracy and the formula complexity, which is a common practice in SR tasks Cavalab (2022). For the baseline methods, in addition to the three types of methods considered in the ground-truth problem set, we also consider decision tree-based methods (see Appendix E.1.1 for details), where the constructed decision trees are treated as a generalized formula tree that uses conditional selection as operators and the number of nodes in the trees are used as the formula complexity.

**Experimental Results** The overall performance of our method and other baselines is

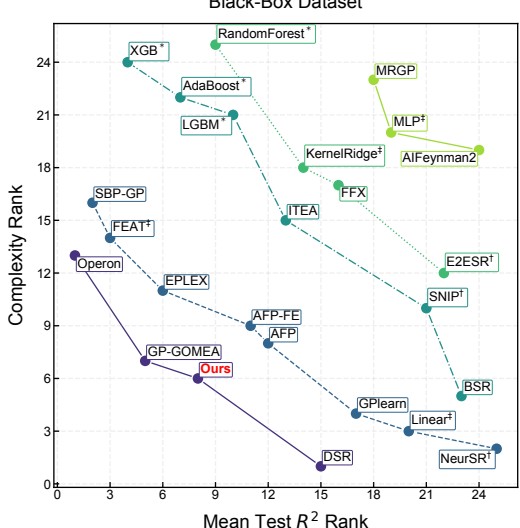

Figure 5: **Pareto fronts on black-box dataset.** †, ‡, and ∗ denote generative, regression, and decision-tree methods, respectively, the others are search methods. The colored lines mark the Pareto front in different ranks, from bottom left (best) to upper right (worst).

shown in Figure 5, where we can see that, our method achieves a higher test set $R^2$ with lower model complexity than other methods and is thus sharing the first rank of the Pareto front with three other search algorithms, that is, Operon (Burlacu et al., 2020), GP-GPMEA (Virgolin et al., 2021), and DSR (Petersen et al., 2021). In contrast, decision tree methods or regression methods usually construct formulas that are too complex and lack interpretability to describe the given data; while generative methods are unable to iteratively optimize the results and thus have low test set accuracy.

## 5.3 MDLFORMER PERFORMANCE

**Experimental Settings** To give an in-depth discussion about the MDLformer's estimation performance, we generate $K = 1024$ pairs of symbolic formulas and numeric data as described in Sec-

Table 2: **Prediction performance of MDLformer.**

| RMSE | $R^2$ | AUC |
|---|---|---|
| 3.9105 | 0.9035 | 0.8859 |

tion 3.2, where the independent variables $x$ are sampled from GMM. Given that the relative relationship of predicted values is more important than their absolute magnitudes when using MDLformer to guide symbolic regression, we consider ranking metrics in addition to the commonly used root mean squared error (RMSE) and coefficient of determination ($R^2$). Particularly, we consider the area under the ROC curve (AUC) (Fawcett, 2006), which assesses the likelihood that the predicted values and true values maintain consistent ordering when randomly selecting pairs (formalized in Appendix E.2.1). As shown in Table 2, the RMSE metric indicates an average error of 3.9, which is only $15\%$ of the average formula length of 25. The $R^2$ and AUC are also close to $1.0$, demonstrating the excellent performance of MDLformer in predicting the complexity of formulas corresponding to the data. We also provide the results of the other three metrics in Appendix E.2.1.

**MDLformer indicates correct search directions.** Here we use a case study to illustrate that the minimum description length (MDL) estimated by MDLformer has the optimal substructure, that is: 1) the subformula of the target formula has a lower MDL than other formulas and 2) the MDL monotonically decreases along the construction route of the target formula $f$. We consider the Feynman III.15.12 formula, a typical example of successful recovery by our method. As shown in Figure 6, we consider the Feynman III.15.12 equation, a typical example that our method successfully recovered (See Appendix E.2.4 for more examples, including both successful and failed). This equation can be obtained from four steps of transformations $T_{1:4}^*$. Specifically, starting from $d_0 \equiv x$, we iteratively operate the transformation $T_i^*$ that eventually maps $d_0$ to $d_4 \equiv y$. For each $d_i$, we estimate its MDL concerning $y$ with the MDLformer and plot the result in the red line, finding the estimated MDL does monotonically decrease as the formula form gets closer to the target formula. For each $d_i$, we also draw the estimated MDL of $d' = T'(d_i)$ for all possible symbolic functions $T'$ as the blue and white sectors. The blue parts show those lower than the correct transformation result $d_{i+1}^* = T_i^*(d_i)$, which can lead to incorrect search paths. However, the proportion of blue parts is quite low, indicating that the correct search direction aligns well with the direction of the steepest MDL decrease.

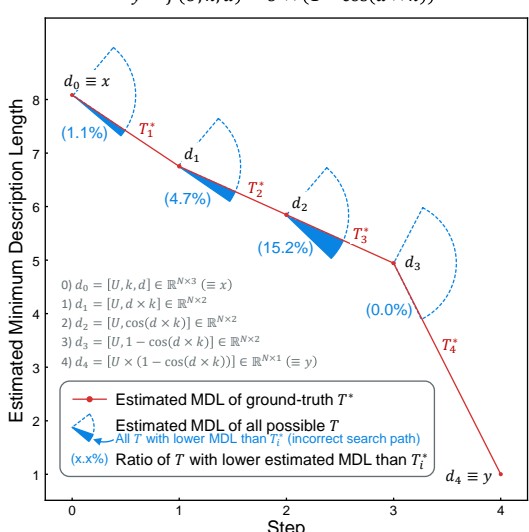

Figure 6: **A case study of Feynman III.15.12 equation.** A series of symbolic functions $T_i^*$ map $d_0 \equiv x$ to $d_4 \equiv y$ iteratively. The red line shows $\mathcal{M}_\Phi(d_i, y)$, the estimated MDL of each $d_i$. The blue and white sectors show the estimated MDL for all possible symbolic functions $d' = T'(d_i)$, where blue sections indicate $T'$ with an estimated MDL lower than $T^*$, which can lead to incorrect search paths. However, the occurrence of these cases, as annotated by the percentage value, is nearly negligible.

**Ablation study on training strategy.** Figure 7a shows the ablation experiment on the training strategy. We considered three strategies to train the MDLformer with the alignment objective $\mathcal{L}_{\text{align}}$ and the prediction objective $\mathcal{L}_{\text{pred}}$: 1) sequentially training for alignment and then prediction (i.e., the training scheme described in Sec 3.3), 2) training for alignment and prediction concurrently, and 3)

direct training for prediction without alignment. A total of 1000 training rounds were performed for each of the three strategies, where for the first method, the first 500 rounds are for alignment and the last 500 rounds are for prediction. Among them, the first two strategies have lower RMSE than the last one, demonstrating that aligning numerical and symbolic spaces improves cross-model prediction, which is consistent with the observation of previous work (Meidani et al., 2023). The Sequential strategy has a lower RMSE than the Concurrent strategy, which may be because it is more difficult to use both objectives at the same time. Figure 7b shows the relationship between the prediction error and the number of input points $N$, demonstrating that more data can help MDL-former predict the minimum description length more accurately. However, a larger $N$ can lead to an increase in inference time, thus we choose $N = 200$ to balance the accuracy and efficiency.

**Consistent predictive capability across task difficulty.** In Figure 7d,e,f we measure the predictive performance of MDLformer on formulas of different difficulty, indicated by 1) the number of variables, 2) the number of unary operators, and 3) the number of binary operators. Although RMSE, consistent with intuition, increases with the formula difficulty. But considering that more difficult formulas can have longer lengths, we additionally plot the ratio of $RMSE$ to the average formula length, $\bar{L}$, in the graph. It can be seen that as the formula difficulty increases, the normalized RMSE remains basically unchanged, indicating that our MDLformer scales well with the formula difficulty.

**Robustness against noise.** In Figure 7c we add feature noise to the data of independent variables: $x \leftarrow x + n$, where $n \sim \mathcal{N}(0, \eta\sigma_x)$ is additive noise with intensity proportional to the standard deviation of $x$, $\sigma_x$. We find that although the RMSE increases with the noise intensity, the ranking metric, AUC, basically remains unchanged as noise increases, suggesting that our method can identify the relative magnitude of MDL even under noisy conditions. Considering that the search process relies solely on the relative magnitude of estimated MDL, this explains the reason for the robustness of our method to noise in Section 5.1.

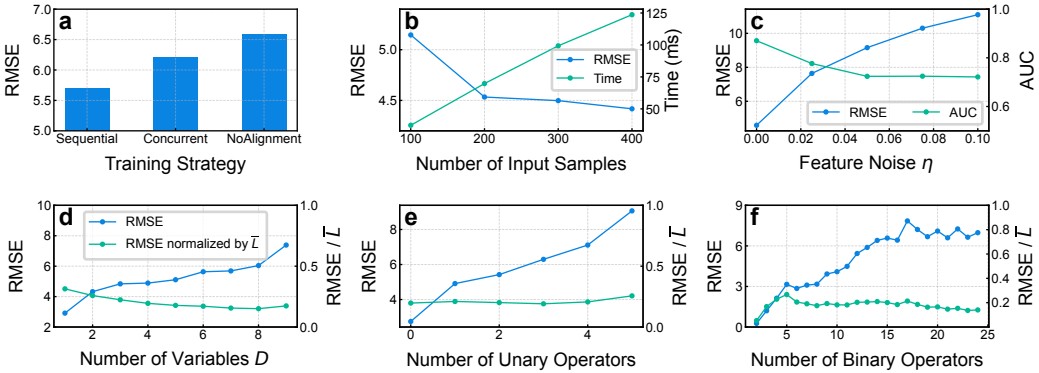

Figure 7: **Ablation Study.** The prediction performance of MDLformer with respect to **a**: trained with three alignment strategies, **b**: number of input pairs $N$, **c**: feature noise $\eta$, **d**: number of variables $D$, **e**: number of unary operators, and **f**: number of binary operators. In **d,e,f** we also plot the RMSE normalized by the average formula length $\bar{L}$.

## 6    DISCUSSION AND CONCLUSION

In this work, we introduce SR4MDL, a symbolic regression approach that optimizes for minimum description length (MDL) rather than prediction error. Leveraging the impressive prediction capabilities of MDLformer, it successfully recovers around 50 formulas across two benchmark datasets comprising 133 problems, outperforming state-of-the-art methods by 43.92%. Additionally, it ranks in the top tier for finding formulas that balance accuracy and complexity on a black-box problem set with 122 problems. While SR4MDL showcases robustness and versatility in searching for correct formulas in a near-optimal direction, it has notable limitations. First, the performance of MDL-former on data with complex relationships needs to be improved. Second, although the increased efficiency caused by the MDLformer makes our method faster than the regression methods, it is still lower than generative methods. Despite these limitations, SR4MDL offers a wide range of capabilities, making it a powerful tool for uncovering symbolic laws that underlie diverse complex systems like city (Zhang et al., 2025), climate (Zhao et al., 2024), ecology (Holland et al., 2002), etc.

ACKNOWLEDGMENTS

This work was supported in part by the National Natural Science Foundation of China under Grant No.U24B20180 and No.62476152. This work is also supported in part by the National Key Research and Development Program of China under Grant No.2020YFA0711403. This work is also supported in part by the Beijing National Research Center for Information Science and Technology (BNRist).

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

## A    RELATIONSHIP WITH RELATED WORK

To better illustrate our innovation, we compare the proposed method with two types of existing methods: 1) heuristic search methods and 2) neural network-based generative methods, as shown in Figure 8. As illustrated in the figure, our method differs from heuristic search methods in terms of its search objective, which shifts from accuracy to the MDL estimated by MDLformer. On the other hand, unlike neural network-based generative methods, where the trained transformer directly generates the symbolic sequence in an end-to-end manner, our proposed method incorporates the MDLformer as a component of the framework, which provides estimated MDL values that serve as search objectives for the search algorithm.

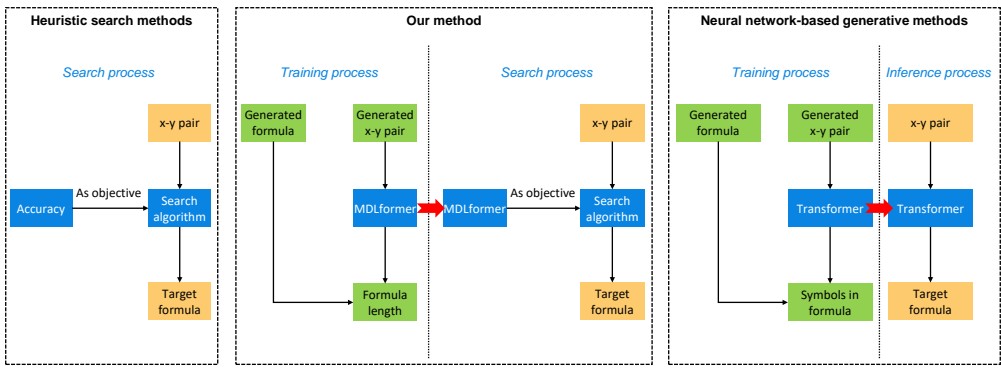

Figure 8: Comparison of heuristic search methods, neural network-based generative methods, and our methods.

Furthermore, many recent studies focus on **neural-symbolic reasoning**, that is, adopting neural networks for symbolic reasoning (van Krieken et al., 2023). These works focus on the problems that, between the input feature $\mathbf{x}$ and target value $\mathbf{y}$, there exists a symbolic "concept" $\mathbf{w}$ that determines $\mathbf{y}$ in a known way $\mathbf{y} = c(\mathbf{w})$. The knowledge of $c$ can thus be used to help predict $\mathbf{y}$ and reason $\mathbf{w}$ behind the input $\mathbf{x}$. For example, if we wanna predict the sum of digits in a series of MNIST images $\mathbf{x}$, we can benefit from the knowledge that 1) each image $x_i$ contains a digit $w_i$ and 2) given all digits $w_i$, the sum $y$ can be determined by $y = \sum_i w_i$. However, despite the similar names, neural-symbolic reasoning is a different task than symbolic regression since no such symbolic concept exists between the input data $(x, y)$ and the target formula $f$ in symbolic regression. This makes neural symbolic reasoning methods unsuitable for symbolic regression tasks.

## B    TECHNICAL DETAILS OF MINIMUM DESCRIPTION LENGTH

The minimum description length (MDL) represents the size of the simplest model used to describe the data(Kolmogorov, 1963). Although it is an abstract and incomputable concept in many fields, it is rather concrete in the field of symbolic formulas. Specifically, for a pair of input-output data $(x, y)$, the minimum number of symbols (i.e., variables, operators, and parameters) required to describe the target formula $y = f(x)$ is its MDL, which is quite easy to calculate when $(x, y)$ is generated by a known formula $f$. For example, $f(x) = x^2 + \sin(x)$ contains five symbols: $[x, \square^2, +, \sin, x]$, so the MDL of data $(x, y = f(x))$ is 5.

One question is how to ensure that the least number of symbols are used to describe $f$, that is, $f$ is in its simplest form. In Section3.2 we simplify the generated formulas with sympy, an open-source Python package for algebra processing, before calculating its length as the training label. The sympy package provides several ways to simplify a formula, such as `expand` that expands a formula, `factor` that factors the formula, `collect` that collects common powers of a term in an expression, etc. We simplify each formula with different methods and choose the shortest simplification result as the simplest form of the formula. Manual checking on 100 simplification results shows that sympy can reach human-level simplification, simplifying most of the formulas to the shortest length that humans can do.

## C  DETAILS OF MDLFORMER

### C.1  MODEL ARCHITECTURE OF MDLFORMER

The MDLformer integrates an embedder, a Transformer encoder, an attention-based pooling layer, and a multilayer perceptron (MLP) as a readout head. The embedder first pad $x \in \mathbb{R}^{N \times D}$ to $\mathbb{R}^{N \times D_{\max}}$, where $D_{\max} = 10$, then concatenate it with $y$ to obtain $(x, y) \in \mathbb{R}^{N \times D_{\max}+1}$. The results are tokenized to triplets of sign, mantissa, and exponent, and are then embedded to $D_i$-dimensional embedding space, forming the $\boldsymbol{E} \in \mathbb{R}^{N \times D_{max}+1 \times 3 \times D_i}$, where $D_i = 64$. To reduce the length of the input sequence of the Transformer encoder, we adopt an MLP with a hidden layer that maps $\boldsymbol{E}$ to $\boldsymbol{E}' = \mathbb{R}^{N \times D_f}$, where $D_f = 512$. Without positional encoding, $\boldsymbol{E}'$ is directly fed into an 8-layer Transformer encoder with 8 heads and 512 hidden units, whose feed-forward layer has a hidden size of 2048. The output of the Transformer encoder, $\boldsymbol{V} \in \mathbb{R}^{N \times D_f}$, is then pooled to $\boldsymbol{n} \in \mathbb{R}^{D_f}$ through the attention-based pooling layer. The $\boldsymbol{n}$ is then fed into the read-out MLP with a hidden layer to obtain the estimated MDL $\mathcal{M}_\Phi(x, y) \in \mathbb{R}$. We clip $N$ to no more than $N_{\max} = 200$ at the input of MDLformer. There are 31.9 million trainable parameters in the MDLformer and 30.7 million trainable parameters in the Symbolic encoder used for the alignment training objective.

### C.2  PRE-TRAINING OF MDLFORMER

We train the MDLformer in a two-step way: First, we use the alignment loss, $\mathcal{L}_{\text{align}}$, to train the MDLformer for 100k steps. Then, we use the prediction loss, $\mathcal{L}_{\text{pred}}$, to train the MDLformer for another 30k steps. During the training, we use a batch size of $1024$, which consumes CUDA memory of around 150G. We use a learning rate of $10^{-3}$ and the noam schedule (Vaswani et al., 2017), where the warmup step is $4000$. During the training, we use a dropout rate of $0.1$. Trained on a machine with 8 A100-SXM4-80GB GPUs and an AMD EPYC 7742 64-Core processor, the training lasted 41 hours, during which 131 million pairs of input data were generated.

## D  DETAILS OF OUR SR METHOD

We compare three different search methods in Table 3:

**Genetic Programming (GP).** GP (Stephens, 2016) maintains a set of candidate formulas, called population. At the initialization step, a random generation algorithm generates initial candidates in a given number as the starting population. In the update step, there are two kinds of functionals to operate on these candidates: crossover and mutation. For the crossover, two candidates are sampled from the population, whose random subformulas are then swapped with each other to obtain the new candidates. For the mutation, a candidate is sampled from the population, whose random subformula is replaced with another formula generated by a formula generator.

**Monte Carlo tree search (MCTS).** MCTS (Sun et al., 2022) maintains a search tree, where each node represents a formula $f_i$ and some numbers, including total rewards $Q_i$ and total counts $N_i$. At the beginning, the search tree contains only an "empty" formula $f = \square$, where $\square$ represents a placeholder to be filled. In the update step, the MCTS algorithm starts from the root node and uses the greedy algorithm to select a formula with the largest upper confidence interval (UCB) that is not in the search tree. Specifically, it considers all possible formulas $f_i$ obtained by filling in a mathematical symbol into a placeholder of current formulas and selects the one with the maximal UCB

$$\text{UCB}_i = \frac{Q_i}{N_i} + \sqrt{\frac{\ln(\sum_i N_i)}{N_i}}, \tag{5}$$

where $\sum_i$ is the sum of all possible formulas. By iteratively repeating this selection until the selected formula $f_i^*$ is not in the search tree, MCTS finds the formula to be added to the candidate set in this round. The placeholders in this formula (if any) are filled with random formulas to create several complete formulas, whose averaged prediction errors in mean square error, $R$, are used to update $Q_i$ for $f_i^*$ and its ancestors:

$$Q_i \leftarrow Q_i + \frac{\eta^{C[f^*]}}{1 + R/\sigma_y^2}, \tag{6}$$

where $\eta = 0.999$, $C[f^*]$ that evaluates the length of $f^*$ is used as a regular term, $\sigma_y$ is the standard deviation of $y$.

**Our Method.** Based on the MCTS algorithm, our method also maintains a search tree. The difference is that each node in the tree $f_i \in \mathcal{F}_{D \times D_i}$ represents $D_i$ subformulas in the target formula, where $D_i$ is no more than 10. For a node $f = \{f^{(1)}, \cdots, f^{(D_i)}\}$, its child node set can be obtained by applying all possible mathematical operators (e.g., $+, \times, \sin$) to all possible combinations of $\{f^{(1)}, \cdots, f^{(D_i)}\} \cup \{x_1, \cdots, x_D\}$ and adding the results to $f$, with or without keeping the operands. We set the root node of the search tree as $f(x) = x \in \mathcal{F}_{D \times D}$. As an example, for a problem with $D = 2$, we initialize the root node of the search tree as $f_{0,1} = \{x_1, x_2\}$, and its child node can be

$f_{1,1} = \{x_1 + x_2, x_1, x_2\}, f_{1,2} = \{x_1 \times x_2, x_1, x_2\}, f_{1,3} = \{\sin x_1, x_1, x_2\}, f_{1,4} = \{\sin x_2, x_1, x_2\},$
$f_{1,5} = \{x_1 + x_2, x_1\}, \cdots ,$
$f_{1,9} = \{x_1 + x_2, x_2\}, \cdots ,$
$f_{1,13} = \{x_1 + x_2\}, \cdots , f_{1,16} = \{\sin x_2\}.$

Starting from the root node, we obtain the node to be added to the search tree in each round by using the greedy algorithm to select the $f_i^*$ with maximal PUCT Silver et al. (2016):

$$\text{PUCT}_i = \frac{Q_i}{N_i} + c_{\text{puct}} \frac{1}{\mathcal{M}_\Phi(f_i(x), y)} \frac{\sqrt{\sum_i N_i}}{N_i + 1}. \tag{7}$$

where $c_{\text{puct}} = 1.41$, the index $i$ denotes all possible formulas that can be obtained by operating an unary or binary mathematical operator on 1 or 2 operands, which can be subformulas contained in the current node, variables, or numeric constants. Apart from these, the other parts are the same as the traditional MCTS approach introduced above.

Table 3: Comparision of existing search regression methods and ours

| Method | Step (Initialize, Update, Select) |
|---|---|
| **GP** | $\{f_i\}_{i \in \text{Population}} := \{\text{Randomly Generated } f\}$
$\{f_i\}_i \leftarrow \{f_i\}_i \cup \{\text{Cross}(f_i, f_j)\}_{i,j} \cup \{\text{Mutate}(f_i)\}_i$
$\{f_i\}_i \leftarrow \text{topk}_{f \in \{f_i\}_i} \|y - f(x)\|^2$ |
| **MCTS** | $\{f_i\}_{i \in \text{MCtree}} := \{\square\}$ (an empty formula without any operators)
$\{f_i\}_i \leftarrow \{f_i\}_i \cup \{f_i^*\}$, where $f_i^*$ has a maximum UCB selected by a greedy algorithm
- |
| **Ours** | $\{f_i\}_{i \in \text{MCtree}} := \{x\}$
$\{f_i\}_i \leftarrow \{f_i\}_i \cup \{f_i^*\}$, $f_i^*$ has a maximum PUCT selected by a greedy algorithm
- |

# E  DETAILED EXPERIMENT RESULTS

## E.1  SYMBOLIC REGRESSION

### E.1.1  BASELINES

In our experiments, we considered a large number of baseline methods, both the ones that come with SRbench and the most recently proposed ones, ensuring the breadth of comparison.

**Search methods.** The vast majority of the methods we compare are search-based algorithms. Among them, genetic programming algorithms are the primary, including GPlearn (Stephens, 2016), AFP & AFP-FE (Schmidt & Lipson, 2010), ITEA (de Franca & Aldeia, 2021), EPLEX (La Cava et al., 2016), MRGP (Arnaldo et al., 2014), SBP-GP (Virgolin et al., 2019), GP-GOMEA (Virgolin et al., 2021), and Operon (Burlacu et al., 2020). In addition to this, we also compare some recent approaches based on reinforcement learning, including SPL that leverages the Monte Carlo tree search (Sun et al., 2022), DSR that relies on the deep reinforcement learning (Petersen et al., 2021), and RSRM that uses a double Q-learning algorithm (Xu et al., 2024). Finally, recent approaches that

combine multiple search methods are also compared, including AI Feynman 2.0 (Udrescu et al., 2020) and BSR (Jin et al., 2020),

**Other methods.** We compare a series of novel methods that leverage large-scale pre-trained Transformer-based neural networks to predict the target formulas directly, including NeurSR (Biggio et al., 2021), E2ESR (Kamienny et al., 2022), and SNIP (Meidani et al., 2023).

**Regression methods.** We also include a neural network-based regression method, FEAT (La Cava et al., 2019), For the experiment on the black box problem set, additional regression methods based on linear regression, MLP fitting, and Kernel Ridge model are included as well.

**Decision tree methods.** For the experiment on the black box problem set, we also consider the XGBoost (Chen & Guestrin, 2016), AdaBoost (Freund & Schapire, 1997), Random Forest (Rigatti, 2017), and LightGBM (Ke et al., 2017) model, which can be considered as a special type of formula with discontinuous selection operators,

### E.1.2 GROUND-TRUTH PROBLEM SET

We provided the detailed experiment results on the ground-truth problem set in Table 4, 5, 6, and 7, finding the recovery rate of our method always outperforms other methods, while its running time is always within a reasonable range. Note that although we provide complexity in these tables, it is not that a smaller value is better, since the gound-truth formula itself has its length. We also provide the Pareto fronts balancing "recovery rate – search time" and "test $R^2$ – complexity" as in Figure 9, finding our method always on the rank 1 of the Pareto front.

### E.1.3 BLACK-BOX PROBLEM SET

We provide the detailed result of the experiments on the black-box problem set as in Table 8, where our method achieves the rank 1 of the Pareto front within a reasonable timeframe, meaning it can balance the accuracy (measured by the test set $R^2$) and simplicity (i.e., formula complexity) better than other baseline methods.

### E.2 PREDICTION PERFORMANCE OF MDLFORMER

### E.2.1 METRICS

In addition to the RMSE, $R^2$, and AUC metrics we provided in the main text, we also considered another three metrics as in Table 9. All metrics illustrate the excellent performance of MDLformer in estimating the value of MDL. The definitions of these metrics are provided as follows:

- **RMSE** evaluates the average difference between the prediction value and the ground truth:

$$\text{RMSE} = \sqrt{\frac{1}{K}\sum_i^K (y_i - \hat{y}_i)^2} \tag{8}$$

- $\boldsymbol{R^2}$ evaluates the strength of the linear relationship between the predicted value and the target value:

$$R^2 = 1 - \frac{\sum_i^K (y_i - \hat{y}_i)^2}{\sum_i^K (y_i - \bar{y})^2} \tag{9}$$

- **AUC** is a ranking metric, which evaluates the likelihood that the predicted values and the true values maintain consistent ordering when randomly selecting pairs:

$$\text{ROC} = \frac{\sum_i^K \sum_{j \neq i}^K \mathbb{I}((y_i - y_j)(\hat{y}_i - \hat{y}_j) > 0)}{K(K-1)}, \tag{10}$$

    where $\mathbb{I}$ is the indicator function.

- $\rho_{\text{Spearman}}$ is another ranking metric, which is defined as

$$\rho_{\text{Spearman}} = 1 - \frac{6\sum_i^K (r_i - \hat{r}_i)^2}{N(N^2-1)}, \tag{11}$$

    where $r_i$ is the rank of $y_i$, ranging from small to large, and the similar as $\hat{r}_i$.

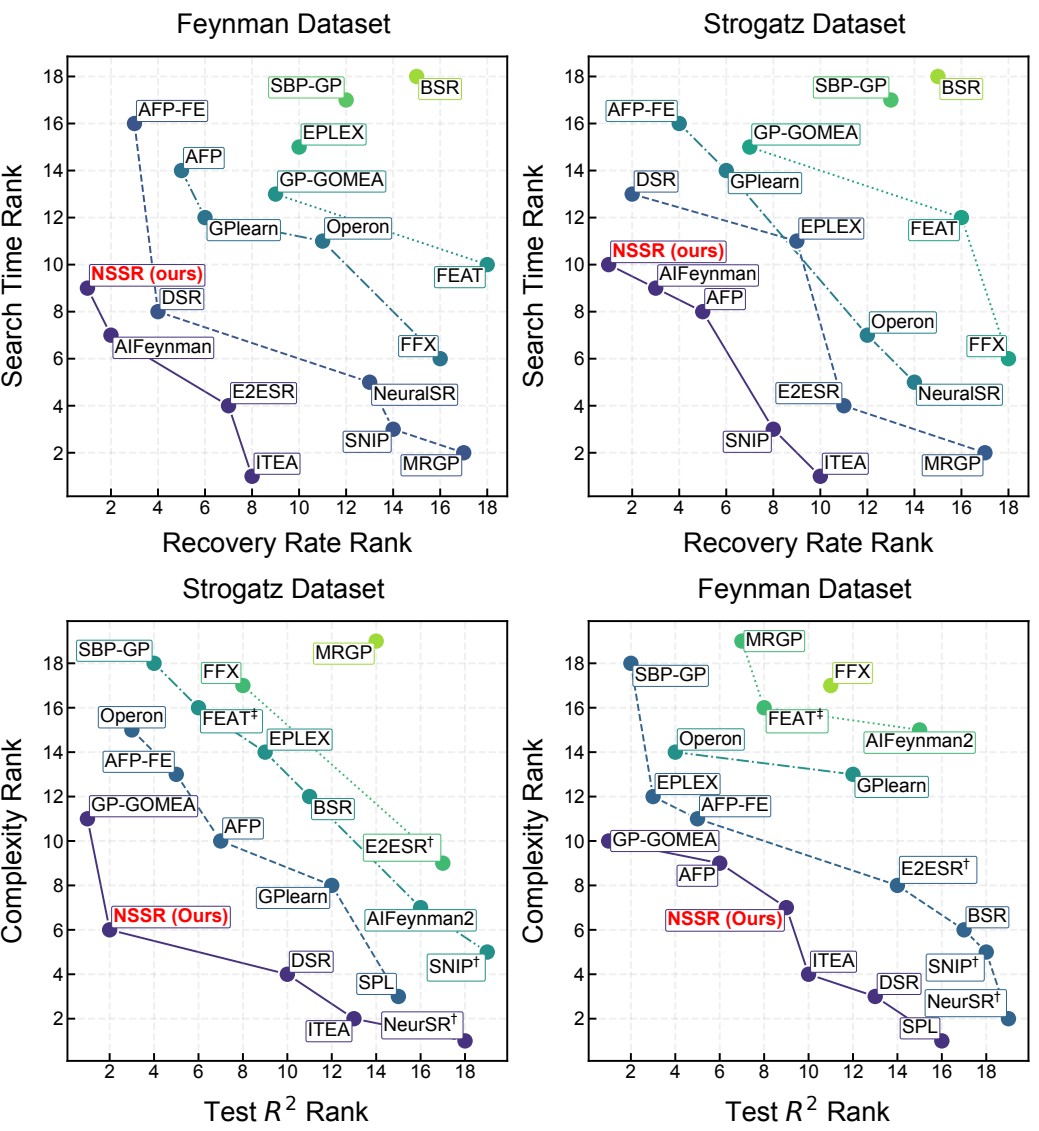

Figure 9: **Pareto fronts balancing "recovery rate – search time" and "test $R^2$ – complexity" on Feynman and Strogatz datasets.**

Table 4: **Complete experimental results at a noise level $\epsilon = 0.0$.** The results include test set $R^2$, recovery rate, search time, and model size of different methods in the Feynman and Strogatz datasets, each experiment is conducted under 10 random seeds. (The same as Tables 5,6,7.)

| Type | Method | Strogatz Dataset ($\epsilon = 0.0$) | | | | Feynman Dataset ($\epsilon = 0.0$) | | | |
|---|---|---|---|---|---|---|---|---|---|
| | | $R^2\uparrow$ | R. Rate$\uparrow$ | Time (s)$\downarrow$ | Complx$\downarrow$ | $R^2\uparrow$ | R. Rate$\uparrow$ | Time (s)$\downarrow$ | Complx$\downarrow$ |
| Regression | FEAT | 0.9210(±0.054) | 0.92%(±3%) | 1135(±970) | 119(±16) | 0.9190(±0.014) | 0.00%(±0%) | 2417(±1737) | 205.3(±13) |
| Generative | NeurSR | 0.5206(±0.029) | 0.00%(±0%) | 14.82(±1.7) | **11.3**(±0.51) | 0.3958(±0.013) | 3.20%(±1%) | 24.18(±2.2) | **13.28**(±0.15) |
| | E2ESR | 0.5341(±0.043) | 6.06%(±5%) | 3.729(±2) | 32.49(±2.9) | 0.8570(±0.013) | 12.90%(±1%) | 4.062(±1.3) | 36.02(±0.88) |
| | SNIP | 0.3995(±0.093) | 7.14%(±0%) | **1.371**(±0.17) | 16.77(±0.59) | 0.6480(±0.037) | 1.68%(±1%) | **1.901**(±0.14) | 24.87(±0.10) |
| Search | GPlearn | 0.7689(±0.067) | 8.93%(±3%) | 1195(±592) | 28.96(±2.8) | 0.8809(±0.027) | 16.41%(±4%) | 3900(±1052) | 72.43(±29) |
| | AFP | 0.9248(±0.041) | 15.18%(±6%) | 192.2(±106) | 38.17(±2.8) | 0.9590(±0.005) | 21.12%(±2%) | 3655(±1999) | 36.87(±2.2) |
| | AFP-FE | 0.9442(±0.045) | 20.00%(±11%) | 11041(±14277) | 46.16(±4.1) | 0.9806(±0.007) | 26.98%(±3%) | 17817(±530) | 39.97(±1.9) |
| | EPLEX | 0.8125(±0.065) | 8.93%(±5%) | 548.2(±258) | 50.09(±2.7) | 0.9869(±0.006) | 12.39%(±2%) | 12771(±4998) | 52.95(±1.3) |
| | SBP-GP | 0.9812(±0.016) | 11.61%(±5%) | 17591(±1765) | 712.2(±39) | 0.9945(±0.001) | 12.72%(±2%) | 28901(±37) | 489.4(±16) |
| | GP-GOMEA | **0.9925**(±0.009) | 29.46%(±10%) | 2760(±1258) | 36.43(±2.4) | **0.9956**(±0.003) | 26.83%(±2%) | 5030(±967) | 34.57(±1.5) |
| | Operon | 0.9878(±0.023) | 11.43%(±6%) | 66.58(±10) | 59.23(±2.1) | 0.9889(±0.006) | 16.55%(±4%) | 2174(±373) | 69.88(±1.8) |
| | SPL | 0.7390(±0.047) | 7.94%(±2%) | 322.1(±180) | 14.55(±2.5) | 0.7073(±0.011) | 10.37%(±1%) | 209(±145) | 12.88(±0.57) |
| | DSR | 0.7602(±0.086) | 19.64%(±6%) | 1858(±2617) | 15.6(±1.7) | 0.8441(±0.091) | 19.72%(±8%) | 1733(±3105) | 14.86(±1) |
| | RSRM | 0.5501(±0.103) | 4.20%(±4%) | 121.9(±36) | 13.09(±2.3) | 0.8003(±0.013) | 16.07%(±2%) | 116.3(±31) | 13.17(±0.47) |
| | AIFeynman2 | 0.6459(±0.039) | 28.24%(±0%) | 762.1(±424) | 22.26(±1.7) | 0.9314(±0.016) | **57.32%**(±1%) | 854.3(±24) | 124.5(±16) |
| | BSR | 0.8455(±0.044) | 0.89%(±3%) | 31380(±23952) | 38.98(±5.4) | 0.6609(±0.018) | 2.48%(±1%) | 29065(±765) | 25.5(±0.23) |
| | Ours | 0.9900(±0.009) | **67.86%**(±10%) | 186.6(±35) | 14.07(±1.9) | 0.9171(±0.005) | 39.92%(±2%) | 467.3(±415) | 23.4(±1.2) |
| | Rank | 2 | 1 | 6 | 3 | 9 | 2 | 6 | 5 |

Table 5: **Complete experimental results at a noise level $\epsilon = 0.001$.**

| Type | Method | Strogatz Dataset ($\epsilon = 0.001$) | | | | Feynman Dataset ($\epsilon = 0.001$) | | | |
|---|---|---|---|---|---|---|---|---|---|
| | | $R^2\uparrow$ | R. Rate↑ | Time (s)↓ | Complx↓ | $R^2\uparrow$ | R. Rate↑ | Time (s)↓ | Complx↓ |
| Regression | FEAT | 0.9244(±0.032) | 0.00%(±0%) | 594.4(±181) | 106.7(±15) | 0.9207(±0.006) | 0.00%(±0%) | 1726(±242) | 196.5(±12) |
| Generative | NeurSR | 0.5219(±0.031) | 0.00%(±0%) | 15.07(±1.7) | **11.41**(±**0.31**) | 0.3979(±0.013) | 3.45%(±1%) | 24.51(±1.7) | 13.24(±0.13) |
| | E2ESR | 0.5105(±0.060) | 0.00%(±0%) | 3.436(±0.75) | 33.83(±4.4) | 0.8585(±0.010) | 11.46%(±2%) | 3.894(±0.98) | 35.85(±1.5) |
| | SNIP | 0.4220(±0.059) | 6.43%(±2%) | **1.344**(±**0.16**) | 16.66(±0.65) | 0.6576(±0.025) | 2.02%(±1%) | **1.926**(±**0.18**) | 24.88(±0.18) |
| Search | GPlearn | 0.7955(±0.067) | 9.29%(±3%) | 913.3(±121) | 29.59(±2.5) | 0.8902(±0.008) | 17.27%(±3%) | 3316(±540) | 60.49(±12) |
| | AFP | 0.9172(±0.052) | 13.57%(±7%) | 143.4(±27) | 38.75(±4.6) | 0.9606(±0.006) | 19.66%(±2%) | 3711(±457) | 39.33(±1.6) |
| | AFP-FE | 0.9447(±0.042) | 15.71%(±8%) | 8108(±584) | 48.74(±3) | 0.9805(±0.007) | 21.90%(±4%) | 26160(±157) | 46.47(±1.2) |
| | EPLEX | 0.8488(±0.053) | 9.29%(±5%) | 416.1(±81) | 49.26(±4.7) | 0.9866(±0.007) | 9.57%(±1%) | 12341(±436) | 56.03(±1.3) |
| | SBP-GP | 0.9879(±0.015) | 0.00%(±0%) | 19596(±1233) | 820.5(±41) | 0.9953(±0.001) | 0.78%(±1%) | 28940(±20) | 574.4(±13) |
| | GP-GOMEA | 0.9914(±0.009) | 7.14%(±7%) | 804(±603) | 41.14(±2.9) | **0.9962**(±**0.001**) | 11.03%(±2%) | 2904(±146) | 45.23(±0.71) |
| | Operon | 0.9843(±0.036) | 5.00%(±3%) | 75.68(±14) | 67.03(±5.5) | 0.9916(±0.005) | 13.19%(±2%) | 2195(±404) | 69.67(±1.6) |
| | SPL | 0.7526(±0.049) | 7.63%(±2%) | 358.8(±211) | 14.4(±2.5) | 0.7073(±0.016) | 10.28%(±2%) | 275.5(±206) | 13.15(±0.44) |
| | DSR | 0.8067(±0.048) | 19.29%(±3%) | 500.8(±317) | 18.66(±2.2) | 0.8764(±0.003) | 19.14%(±1%) | 830.1(±282) | 16.04(±0.31) |
| | RSRM | 0.5447(±0.105) | 6.06%(±6%) | 129.7(±8.1) | 12.23(±0.65) | 0.8104(±0.025) | 17.18%(±2%) | 128.2(±3.8) | **13.04**(±**0.43**) |
| | AIFeynman2 | 0.6855(±0.091) | 22.14%(±5%) | 84.19(±74) | 25.64(±3) | 0.9177(±0.008) | 33.67%(±4%) | 638(±21) | 130.6(±17) |
| | BSR | 0.8224(±0.121) | 0.71%(±2%) | 24299(±3478) | 37.68(±2.2) | 0.6538(±0.023) | 0.60%(±1%) | 30255(±4770) | 25.85(±0.57) |
| | Ours | **0.9965**(±**0.004**) | **73.24%**(±9%) | 171.5(±30) | 21.38(±1.4) | 0.9079(±0.008) | **34.22%**(±2%) | 428.9(±261) | 32.35(±1.1) |
| | Rank | 1 | 1 | 8 | 6 | 9 | 1 | 6 | 7 |

Table 6: **Complete experimental results at a noise level $\epsilon = 0.01$.**

| Type | Method | Strogatz Dataset ($\epsilon = 0.01$) | | | | Feynman Dataset ($\epsilon = 0.01$) | | | |
|---|---|---|---|---|---|---|---|---|---|
| | | $R^2$↑ | R. Rate↑ | Time (s)↓ | Complx↓ | $R^2$↑ | R. Rate↑ | Time (s)↓ | Complx↓ |
| Regression | FEAT | 0.9244(±0.043) | 0.00%(±0%) | 472.9(±90) | 95.61(±16) | 0.9212(±0.010) | 0.00%(±0%) | 1464(±365) | 167.1(±6.5) |
| Generative | NeurSR | 0.5179(±0.042) | 6.43%(±2%) | 15.48(±1.4) | **11.63**(±0.34) | 0.3942(±0.011) | 3.11%(±1%) | 24.62(±1.7) | 13.26(±0.12) |
| | E2ESR | 0.5031(±0.034) | 1.28%(±3%) | 3.392(±0.72) | 35.94(±1.8) | 0.8345(±0.007) | 9.72%(±2%) | 4.274(±0.58) | 40.07(±0.90) |
| | SNIP | 0.4562(±0.052) | 7.14%(±0%) | **1.418**(±0.22) | 18.1(±1.3) | 0.6550(±0.029) | 2.61%(±2%) | **2.19**(±0.23) | 26.96(±0.40) |
| Search | GPlearn | 0.7956(±0.059) | 9.29%(±5%) | 907.4(±110) | 30.59(±4.3) | 0.8890(±0.009) | 16.99%(±3%) | 3351(±437) | 60.07(±19) |
| | AFP | 0.9153(±0.053) | 11.43%(±6%) | 152.2(±15) | 38.62(±7.1) | 0.9610(±0.005) | 16.90%(±3%) | 4090(±758) | 40.86(±1.1) |
| | AFP-FE | 0.9582(±0.041) | 13.57%(±8%) | 8898(±579) | 48.8(±5.4) | 0.9819(±0.008) | 20.78%(±4%) | 27763(±297) | 46.92(±2.3) |
| | EPLEX | 0.8562(±0.040) | 4.29%(±4%) | 437.6(±64) | 53.07(±3.8) | 0.9910(±0.002) | 8.71%(±2%) | 11043(±718) | 54(±0.68) |
| | SBP-GP | 0.9813(±0.015) | 0.00%(±0%) | 20783(±776) | 850.9(±34) | 0.9950(±0.001) | 0.00%(±0%) | 28954(±15) | 595.5(±12) |
| | GP-GOMEA | 0.9783(±0.029) | 1.43%(±3%) | 765.6(±1005) | 42.64(±4.5) | **0.9967**(±0.001) | 5.09%(±1%) | 3020(±360) | 44.67(±1.1) |
| | Operon | **0.9829**(±0.031) | 0.71%(±2%) | 94.92(±15) | 81.68(±1.2) | 0.9878(±0.010) | 2.07%(±1%) | 3165(±549) | 87.96(±1.3) |
| | SPL | 0.7388(±0.060) | 7.91%(±4%) | 413.3(±295) | 14.71(±1.8) | 0.7133(±0.007) | 11.24%(±2%) | 295.1(±175) | 13.43(±0.58) |
| | DSR | 0.8199(±0.055) | 18.57%(±4%) | 492.9(±287) | 18.51(±1.2) | 0.8782(±0.004) | 18.97%(±1%) | 929.8(±422) | 16.2(±0.41) |
| | RSRM | 0.5969(±0.077) | 4.31%(±5%) | 142.2(±21) | 14.22(±1.5) | 0.8092(±0.015) | 15.00%(±3%) | 131.6(±14) | **12.97**(±0.34) |
| | AIFeynman2 | 0.7753(±0.047) | 9.29%(±8%) | 85.17(±75) | 32.41(±4.1) | 0.8732(±0.021) | 13.86%(±4%) | 629.4(±5.9) | 155.2(±8) |
| | BSR | 0.8127(±0.070) | 0.00%(±0%) | 23622(±554) | 38.74(±2.6) | 0.6734(±0.018) | 0.09%(±0%) | 30411(±4711) | 28.03(±0.49) |
| | Ours | 0.9718(±0.057) | **62.86%**(±6%) | 505.1(±34) | 20.31(±0.81) | 0.9140(±0.009) | **31.97%**(±2%) | 844.3(±561) | 31.15(±1.5) |
| | Rank | 4 | 1 | 12 | 6 | 8 | 1 | 7 | 7 |

Table 7: **Complete experimental results at a noise level $\epsilon = 0.1$.**

| Type | Method | Strogatz Dataset ($\epsilon = 0.1$) | | | | Feynman Dataset ($\epsilon = 0.1$) | | | |
|---|---|---|---|---|---|---|---|---|---|
| | | $R^2$↑ | R. Rate↑ | Time (s)↓ | Complx↓ | $R^2$↑ | R. Rate↑ | Time (s)↓ | Complx↓ |
| Regression | FEAT | 0.9228(±0.027) | 0.00%(±0%) | 446.9(±73) | 84.2(±15) | 0.9195(±0.006) | 0.00%(±0%) | 777.9(±102) | 99.48(±5.6) |
| Generative | NeurSR | 0.5054(±0.058) | 0.71%(±2%) | 17.47(±1.6) | **12.74**(±0.37) | 0.3823(±0.020) | 0.00%(±0%) | 25.81(±1.7) | 13.54(±0.16) |
| | E2ESR | 0.5152(±0.038) | 7.14%(±5%) | 5.621(±4.9) | 38.49(±2.7) | 0.7714(±0.014) | 7.01%(±1%) | 6.183(±0.83) | 44.09(±0.61) |
| | SNIP | 0.5536(±0.057) | 6.43%(±4%) | **1.695**(±0.23) | 22.55(±1.2) | 0.6669(±0.020) | 0.08%(±0%) | **2.768**(±0.20) | 30.8(±0.32) |
| | GPlearn | 0.8228(±0.052) | 9.29%(±3%) | 894.6(±108) | 25.84(±2.9) | 0.8911(±0.007) | 16.80%(±2%) | 2938(±543) | 48.83(±9.5) |
| | AFP | 0.9110(±0.065) | 4.29%(±4%) | 161.4(±28) | 44.44(±5.3) | 0.9577(±0.007) | 13.10%(±2%) | 3886(±341) | 40.79(±1.5) |
| | AFP-FE | 0.9496(±0.037) | 2.14%(±3%) | 10082(±565) | 50.96(±4) | 0.9826(±0.005) | 13.53%(±4%) | 28812(±0.51) | 48.87(±1.4) |
| | EPLEX | 0.8822(±0.078) | 2.14%(±5%) | 405.8(±43) | 53.46(±2.3) | 0.9901(±0.004) | 10.17%(±3%) | 10283(±532) | 45.62(±1.3) |
| | SBP-GP | 0.9323(±0.050) | 0.00%(±0%) | 21886(±782) | 901.2(±34) | 0.9905(±0.007) | 0.00%(±0%) | 28932(±83) | 621.9(±6.6) |
| | GP-GOMEA | 0.9668(±0.031) | 0.00%(±0%) | 402.9(±802) | 43.71(±2.1) | **0.9957**(±0.003) | 1.64%(±1%) | 3186(±454) | 46.43(±0.91) |
| | Operon | 0.9380(±0.042) | 0.00%(±0%) | 97.13(±15) | 83.44(±1.2) | 0.9847(±0.008) | 0.09%(±0%) | 3090(±330) | 89.23(±0.65) |
| Search | SPL | 0.7715(±0.044) | 9.09%(±3%) | 355.3(±59) | 13.91(±1.6) | 0.7109(±0.013) | 10.00%(±2%) | 270(±168) | 13.54(±0.50) |
| | DSR | 0.8086(±0.047) | 15.00%(±5%) | 500(±300) | 18.51(±2.2) | 0.8779(±0.002) | 16.81%(±2%) | 814.9(±197) | 16.03(±0.46) |
| | RSRM | 0.5553(±0.057) | 3.81%(±5%) | 138.3(±6.4) | 13.54(±1.1) | 0.8104(±0.016) | 13.98%(±2%) | 133.8(±6) | **12.81**(±0.41) |
| | AIFeynman2 | 0.3170(±0.082) | 2.16%(±4%) | 65.97(±19) | 23.53(±4.6) | 0.2248(±nan) | 0.83%(±nan%) | 710.7(±nan) | 176.6(±nan) |
| | BSR | 0.7190(±0.076) | 0.00%(±0%) | 23292(±510) | 49.54(±4.9) | 0.6567(±0.024) | 0.00%(±0%) | 32497(±7914) | 28.77(±1.1) |
| | Ours | **0.9686**(±0.057) | **62.86%**(±14%) | 522.6(±43) | 20.5(±0.62) | 0.9097(±0.009) | **28.79%**(±1%) | 962.9(±918) | 30.86(±1.4) |
| | Rank | 1 | 1 | 13 | 5 | 8 | 1 | 9 | 7 |

Table 8: **Complete experimental results of black box dataset.**

| Pareto Rank | Method | Test $R^2$ ↑ | Complexity ↓ | Time (s) ↓ |
|---|---|---|---|---|
| **1** | Operon | 0.7945 | 65.69 | 2974 |
| | GP-GOMEA | 0.7381 | 30.27 | 9636 |
| | **NSSR (Ours)** | 0.6258 | 29.88 | 541.7 |
| | DSR | 0.5625 | 9.465 | 36852 |
| **2** | SBP-GP | 0.7869 | 634 | 149344 |
| | FEAT[‡] | 0.7621 | 82.49 | 6432 |
| | EPLEX | 0.7372 | 53.14 | 15796 |
| | AFP-FE | 0.6400 | 36.04 | 6184 |
| | AFP | 0.6333 | 34.89 | 6033 |
| | GPlearn | 0.5390 | 19.06 | 24254 |
| | Linear[‡] | 0.4437 | 17.4 | 0.2447 |
| | NeurSR[†] | 0.1228 | 13.33 | 11.7 |
| **3** | XGB[*] | 0.7496 | 20186 | 236 |
| | AdaBoost[*] | 0.6939 | 9481 | 65.12 |
| | LGBM[*] | 0.6410 | 5734 | 29.9 |
| | ITEA | 0.6295 | 116.7 | 12183 |
| | SNIP[†] | 0.3335 | 38.91 | 3.286 |
| | BSR | 0.2725 | 22.52 | 59822 |
| **4** | RandomForest[*] | 0.6615 | 1517178 | 120.4 |
| | KernelRidge[‡] | 0.5952 | 1824 | 39.19 |
| | FFX | 0.5575 | 1562 | 244.3 |
| | E2ESR[†] | 0.3612 | 61.09 | 7.101 |
| **5** | MRGP | 0.5300 | 10802 | 165007 |
| | MLP[‡] | 0.5238 | 3882 | 30.49 |
| | AIFeynman2 | 0.2110 | 2240 | 86854 |

- $\rho_{\text{Pearson}}$ , or the correlation coefficient, is a widely used metric for estimating the degree of correlation between two variables, which is defined as

$$\rho_{\text{Pearson}} = \frac{\text{cov}(y, \hat{y})}{\sigma_y \sigma_{\hat{y}}} = \frac{\sum_i^K (y_i - \bar{y})(\hat{y}_i - \bar{\hat{y}}_i)}{\sqrt{\sum_i^K (y_i - \bar{y})}\sqrt{\sum_i^K (\hat{y}_i - \bar{\hat{y}}_i)}} \tag{12}$$

- **NDCG$_R$** is a ranking metric that is defined as

$$\text{NDCG}_R = \frac{\sum_r^R (2^{y_{\hat{i}_r}} - 1)/\log_2(r+1)}{\sum_r^R (2^{y_r} - 1)/\log_2(r+1)}, \tag{13}$$

where $y_r$ is the $r$-th smallest value in $y$, $\hat{i}_r$ is the index of the $r$-th smallest value in $\hat{y}$, where we chose $R = 5$.

| RMSE | $R^2$ | AUC | $\rho_{\text{Spearman}}$ | $\rho_{\text{Pearson}}$ | NDCG$_5$ |
|------|-------|-----|--------------|-------------|--------|
| 3.9105 | 0.9035 | 0.8859 | 0.9542 | 0.9506 | 0.7264 |

Table 9: **Detailed Prediction performance of MDLformer.**

### E.2.2 ABLATION STUDY

We provide the AUC metric of the ablation study as in Figure 10, where we can find similar conclusions as in the main text. That is, 1) the sequential training strategy outperforms the other two strategies, 2) the performance of MDL estimation increases as the number of input samples increases, and 3) the RMSE performance decreases as the feature noise gets stronger while the AUC performance keeps robust.

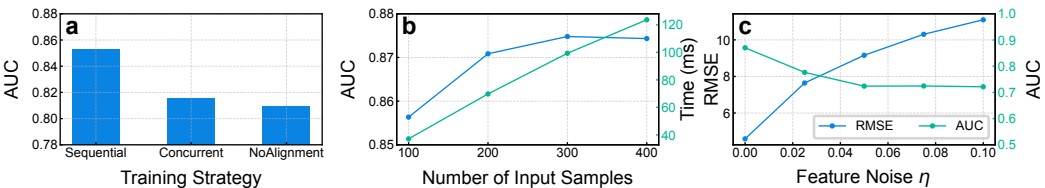

Figure 10: **AUC metric in ablation study.**

In Section 3.2, we used a variety of methods to increase the diversity of generated data, including sampling data dimensions $D$, randomizing formula length $L$, using Gaussian mixture models to sample data distribution $x \sim \text{GMM}$, etc. These designs enhance the diversity of the generated training data, and thus ensure the generalizability of our method on specific symbolic regression tasks. To test the effectiveness of these designs, in Figure11 we test the recovery rate of our SR4MDL method guided by the MDLformer trained with these designed ablated. We conduct experiments on the Strogatz dataset, with noise level $\epsilon = 0$ and 10 random seeds. We considered three kinds of ablation: 1) Data dimensions that replace the random dimension $D \sim \mathcal{U}\{1..10\}$ by a constant dimension $D = 2$; 2) formula lengths that reduce the maximal formula length $L_{\max}$ from 50 to 10; and 3) data distribution that sample data $x$ from $\mathcal{U}(0, 1)$ rather than a more diverse Gaussian mixture model (GMM). As shown in the figure, all ablations result in performance degradation, demonstrating the effectiveness of our design.

### E.2.3 PERFORMANCE WITH RESPECT TO TRAINING SAMPLE SIZE

To study the impact of training sample size on MDLformer's predictive performance and compare it with existing neural network-based generative symbolic regression methods, in Figure fig:evaluateMDLformer we plotted its prediction performance with respect to the number of training samples, as well as the corresponding recovery rates on Strogatz dataset (noise level $\epsilon = 0$,

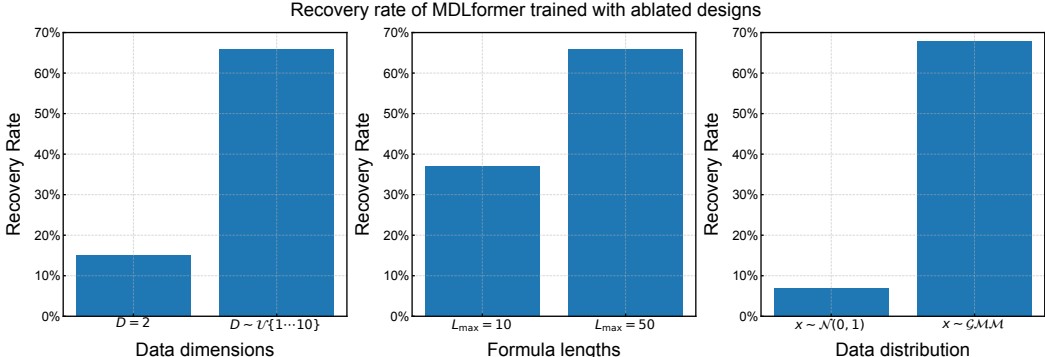

Figure 11: **The recovery rate of MDLformer trained with ablated designs**. We test the recovery rate of our SR4MDL method guided by the MDLformer trained with ablated designs, including data dimensions, formula lengths, and data distribution. The experiments are conducted on the Strogatz dataset, with noise level $\epsilon = 0$ and 10 random seeds.

on 10 random seeds). The prediction performance is measured by both the RMSE (blue line) and the AUC (green line). Since our training is divided into two stages, where in the first stage MDL-former is trained to align the numeric embedding space with the symbolic embedding space but not learn to predict MDL, we only plot the performance curve of the second phase. At the beginning of the second stage, the AUC is about 0.5 and the RMSE is about 25 (i.e., the average length of the formulas in the training set), indicating that the MDLformer does not have the ability to predict the MDL at this point.

As shown in the figure, the trend of RMSE-measured and AUC-measured prediction performance, as well as the recovery rate, are basically the same during the training process: The model performance and recovery rate improve sharply when the training sample size in the second stage reaches $10^6$ (in total of 32 million) and improve again when the training sample size reaches $10^7$ (in total of 41 million). This suggests that MDLformer requires sufficient data to accurately estimate MDL, highlighting the importance of large-scale training corpus. Furthermore, when the data size matches that used in E2ESR (Kamienny et al., 2022) (38.4 million), SNIP (Meidani et al., 2023) (56.3 million), and NeurSR (Biggio et al., 2021) (100 million), the recovery rates reach 42%, 66%, and 67%, respectively, which is substantially higher than those of the baseline methods (¡10%). This demonstrates that the advantage of our approach lies in shifting the search objective from accuracy to MDL instead of the training data volume. Finally, the model begins to converge at a data size of 60 million (in total, the same as below), and the performance does not differ much until 131 million. This indicates that further improvements in future work may require optimizing the data generation process rather than simply increasing the data volume.

### E.2.4 CASE STUDY

In this part, we provide more case studies to demonstrate the optimal substructure in MDLformer's estimation results and discuss how these results determine the success or failure of the SR4MDL algorithm it guides.

In Figure 13 we provide a typical example in the Strogatz dataset – the Gilder-2 problem, where our method efficiently recovers the target formula in a few seconds. As shown in the figure, as the transformation $T_i^*$ iteratively maps $d_0 \equiv x$ to $d_3 \equiv y$, the MDL of $(d_i, y)$ estimated by MDLformer monotonically decreases. At each point of $d_i$, all possible symbolic transformations $T$ lead to an estimated MDL $\mathcal{M}_\Phi(T(d_i), y)$ higher than the correct transformation $\mathcal{M}_\Phi(T_{i+1}^*(d_i), y)$, suggesting that there is no incorrect search path. This explains why our method only takes a few seconds to recover this formula and achieves a significant recovery rate improvement on the Strogatz dataset.

In Figure 14 we further provide two examples on the Feynman dataset, illustrating the performance of our method on the problem of successfully discovering the target formula (left) and on the problem of failing to discover the target formula (right), respectively. As shown in the left plot, for

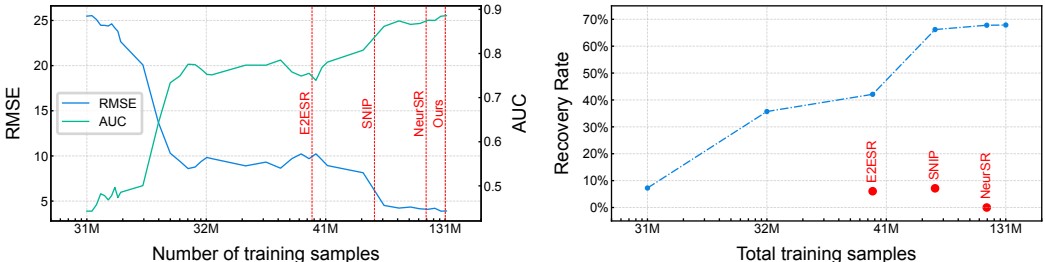

Figure 12: **Prediction performance of MDLformer in different numbers of training samples** The figure only shows the second stage of the training process. 31M is the number of samples used in the first stage of alignment training. For the convenience of comparison, we also mark the number of training samples used by recent works, including NeurSR which uses 100 million samples(Biggio et al., 2021), E2ESR which uses 38.4 million samples (Kamienny et al., 2022), and SNIP which uses 56.3 million samples(Meidani et al., 2023).

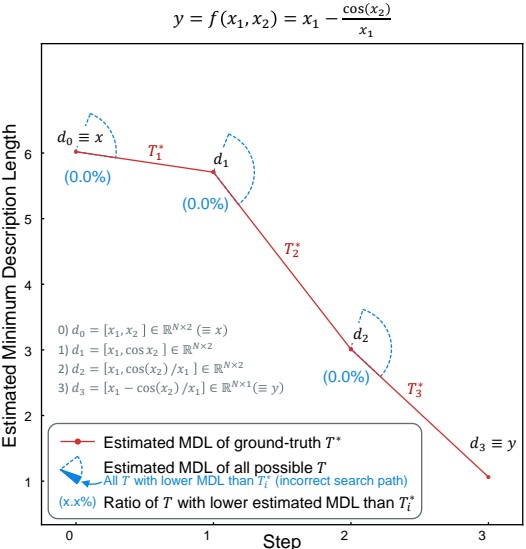

Figure 13: **A case study of the Strogatz dataset.** Here we consider the Gilde-2 problem, a typical example where our method successfully recovers the target formula in a few seconds.

Feynman I.18.4 where our method successfully recovered the target formula, the estimated minimum description length (MDL) monotonically decreases as the transformation gets closer to the target formula (shown by the red line), and the transformations other than the correct $T_i^*$ result in a larger MDL (shown by the white sectors). Transformations resulting in MDL lower than correct transformations can lead to incorrect search paths (shown by the blue sectors). However, there are only a few incorrect transformations (annotated by the percentages), explaining the successful recovery for this problem.

On the other hand, for the Feynman III.10.19 problem shown in the right plot of Figure 14, where our method failed to recover the target formula, the estimated MDL does not decrease monotonically. Specifically, it fails to notice that $T_1^* : B_z \mapsto B_z^2$ and $T_6^* : B_x^2 + B_y^2 + B_z^2 \mapsto \sqrt{B_x^2 + B_y^2 + B_z^2}$ are parts of the target formula. This may be because the square and root operations significantly change the numeric distribution in $d_i$, leading to a decrease in MDLformer's predictive performance. For the same reason, at steps 1, 2, 3 (square operations), and 6 (root operation), a large number of transformations ($13.0\% \sim 66.9\%$) result in a lower MDL than the correct transformation, leading to many incorrect search paths. This explains the failure of our method on this problem and suggests that to further enhance our method, we have to improve its robustness to these non-linear operators that significantly influence numeric distribution.

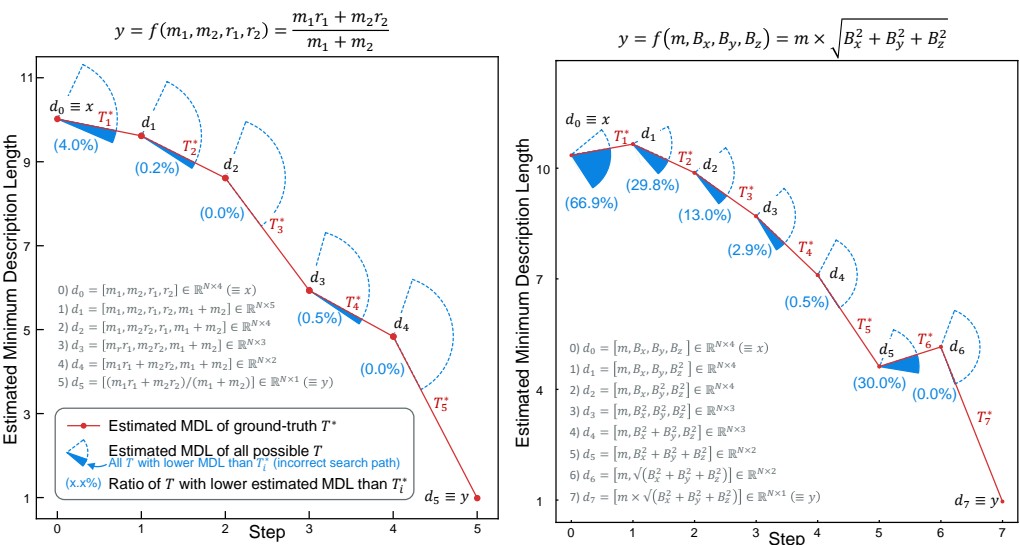

Figure 14: **A case study of success and failure examples in the Feynman dataset.** Here we consider the Feynman I.18.4 (left) and Feynman III.10.19 (right) problems, two typical examples of successful and unsuccessful recovery by our methods, respectively.

