# OpenReview forum: "Symbolic regression via MDLformer-guided search: from minimizing prediction error to minimizing description length"
_ICLR.cc/2025/Conference — ICLR 2025 Poster_

### Official Review · Reviewer_s6CS · 2024-10-30

**Soundness:** 3
**Presentation:** 2
**Contribution:** 3
**Rating:** 6
**Confidence:** 3

**Summary:**

This paper addresses a fundamental challenge in symbolic regression: the traditional search objective, namely prediction error, is non-monotonic, which limits the effectiveness of the search process. The authors propose a novel approach by training a neural network called MDLformer to tackle this. This model introduces a new objective function to efficiently guide the formula search. The experimental results demonstrate that the proposed method significantly outperforms existing approaches.

**Strengths:**

- The overall idea of this paper is intuitive and reasonable. The traditional search objective, i.e., prediction error, cannot effectively capture the distance between the target and current formulas.
- The experimental results are promising. The proposed method significantly outperforms existing methods, demonstrating that a more sophisticated objective can help improve the recovery rate.

**Weaknesses:**

- The evaluation dataset is limited. Unlike existing studies, this paper aims to introduce a neural network to guide the search process. As a result, it is crucial to comprehensively assess whether this model can generalize well on other tasks. However, the two datasets used in this study are relatively small. For instance, the Strogatz dataset only includes 14 problems, which is insufficient to effectively capture the performance of each method. Therefore, incorporating additional datasets [1] is necessary to thoroughly evaluate the proposed approach.
- The proposed MDLformer model requires an in-depth discussion. Firstly, the method generates synthetic data to train the model. How do the authors ensure that this synthetic data generalizes effectively across various downstream tasks? Secondly, with 131 million data points generated, how does the model’s performance change as the data scale increases, and does it eventually reach performance convergence?
- The technical section should be clear and self-contained. Although the concept of Minimum Description Length (MDL) originates from existing studies, further clarity is needed on its technical details, such as how it is computed in the proposed method. At present, the technical content is somewhat difficult to follow.

[1] A Comparison of Recent Algorithms for Symbolic Regression to Genetic Programming. Preprint.

**Questions:**

- The data contamination problem should be further discussed. The authors generate 131 million data points, which poses a potential risk of overlap between training data and testing data. Does the generated data introduce contamination problems, and how can this problem be avoided in the data generation process?
- Some related studies should be discussed. Many recent studies [1] focus on adopting neural networks for symbolic reasoning. The authors should provide a brief discussion about the differences and advantages of the proposed method.

[1] A-NeSI: A Scalable Approximate Method for Probabilistic Neurosymbolic Inference. NeurIPS 2023.

---

> ### Author Response · Authors · 2024-11-19
> **Response to Reviewer s6CS (part1)**
>
> Thanks for your feedback. We have summarized your comments into five questions. We hope the following responses can address your concerns.
>
> **Q1: Limited evaluation dataset and data contamination problems.**
>
> > Weakness 1: The evaluation dataset is limited. Unlike existing studies, this paper aims to introduce a neural network to guide the search process. As a result, it is crucial to comprehensively assess whether this model can generalize well on other tasks. However, the two datasets used in this study are relatively small. For instance, the Strogatz dataset only includes 14 problems, which is insufficient to effectively capture the performance of each method. Therefore, incorporating additional datasets [1] is necessary to thoroughly evaluate the proposed approach.
> > [1] A Comparison of Recent Algorithms for Symbolic Regression to Genetic Programming. Preprint.
>
> **Response:** **First**, in addition to the two datasets including **133** ground-truth problems you mentioned, we also evaluate our method on **122** black-box problems as described in Section 5.2. This is much more sufficient than the benchmark in paper [1] (https://arxiv.org/abs/2406.03585), which contains only **9** black-box problems, and 2 of them are already contained in the 122 problems we used in Section 5.2 (Although the author claims that they are not yet contained in SRBench, we found that `Nikuradse 1 & 2` problems have been contained by SRBench already).
>
> **Secondly**, just as Reviewer 1vTX said, our evaluation dataset follows a standard convention that the related work follows. Here we list the size of the validation dataset used by related work in recent years, including both neural network-based methods and non-neural network methods, where our evaluation experiment used the largest number of problems.
> * Ours: **255** problems
> * RSRM *(Xu, ICLR’24)*: **58** problems
> * SNIP *(Meidani, ICLR’24)*: **190** problems
> * SPL *(Sun, ICLR’23)*: **30** problems
> * E2ESR *(Kamienny, NeurIPS’22)*: **190** problems (and 100k problems generated in the same way as the training data, which aims to test the in-domain performance, so we did not count it)
> * DGSR *(Kamienny, ICML'23)*: **190** problems (and 1k generated problems)
> * NeurSR *(Biggio, PMLR’21)*: **64** problems (and 200 generated problems)
> * DSR *(Petersen, ICLR'21)*: **37** problems
>
> **Finally**, we conduct experiments on the problems introduced in the paper [1]. We looked over [1] and its references, finding only three accessible datasets, whose experimental results are listed as follows:
>
> | Problem | Method | test-set NMSE | Formula length | Time (s) |
> | :-: | :-: | :-: | :-: | :-: |
> | Chemical Comp. | **Ours** | *0.209* | 18 | 1661 |
> |  | HeuristicLab | **0.204** | - | 2975 |
> |  | Operon | 0.270 | - | 29 |
> |  | E2E | 1.229 | - | 92 |
> |  | SciMED AutoML | 0.448 | - | 7560 |
> |  | SciMED GA-SR | 1.124 | - | 600 |
> | Nikuradse 1 | **Ours** | **0.007** | 22 | 117 |
> |  | HeuristicLab | 0.056 | - | 578 |
> |  | Operon | *0.054* | - | 5 |
> |  | E2E | 0.905 | - | 35 |
> |  | SciMED AutoML | 0.734 | - | 716 |
> |  | SciMED GA-SR | 1.005 | - | 600 |
> | Nikuradse 2 | **Ours** | *0.020* | 20 | 176 |
> |  | HeuristicLab | **0.019** | - | 282 |
> |  | Operon | 0.021 | - | 6 |
> |  | E2E | 0.129 | - | 50 |
> |  | SciMED AutoML | 0.020 | - | 1778 |
> |  | SciMED GA-SR | 1.005 | - | 600 |
>
> Across all three problems, our method achieves the best or second-best test set errors (NMSE). For `Chemical Comp.` and `Nikuradse 2` that HeuristicLab outperformed us, we achieved very close fitting accuracy (2% and 5% difference) in much less time (44% and 38% less).
>
> Note that the original article does not provide the formula lengths of the results, which is an important metric for evaluating the formula's interpretability. This makes it impossible for us to compare the Pareto front between fitting error and formula length, as is usually done in symbolic regression methods.
>
>
> **Q2: Does the generated data introduce contamination problems, and how can this problem be avoided in the data generation process?**
>
> > Question 1: The data contamination problem should be further discussed. The authors generate 131 million data points, which poses a potential risk of overlap between training data and testing data. Does the generated data introduce contamination problems, and how can this problem be avoided in the data generation process?
>
> **Response:** The generated data does not introduce contamination problems in principle. Different from existing neural network-based methods that learn *symbols in formulas* $f$ with observational data $(x,y)$ as input, in our method, the MDLformer learns to predict *lengths of formulas* $C[f] \in \mathbb{N}$ with $(x,y)$ as input. Therefore, the trained MDLformer can only estimate the formula length to provide the search objective, but cannot cheat by memorizing and reproducing the formulas in the training set -- since it has never seen the formulas in the training set but only their lengths.

---

> ### Author Response · Authors · 2024-11-19
> **Response to Reviewer s6CS (part2)**
>
> **Q3: How to ensure that the synthetic data generalizes across various downstream tasks? And how does the model’s performance change and eventually converge as the data scale increases?**
>
> > Weakness 2: The proposed MDLformer model requires an in-depth discussion. Firstly, the method generates synthetic data to train the model. How do the authors ensure that this synthetic data generalizes effectively across various downstream tasks? Secondly, with 131 million data points generated, how does the model’s performance change as the data scale increases, and does it eventually reach performance convergence?
>
> **Response:** **For the first question on generalizability**, our design in Section 3.2 generates training data with various problem dimensions, formula lengths, formula forms, and numeric distribution, guaranteeing the diversity of the training data and thus ensuring the generalizability of our model. We have added ablation experiments in **Appendix E.2.2** to demonstrate how these designs effectively help our MDLformer generalize to downstream tasks, where you can see reducing the diversity of problem dimensions, formula forms, and data distribution in training data all lead to a significant decrease in the recovery rate.
>
> **For the second question on the converge process**, we provide MDLformer's prediction performance with respect to the number of training samples in **Appendix E.2.3**, where you can see: **First**, the trend of MDLformer's prediction performance and the recovery rate on downstream search tasks are synchronized during the training process. **Secondly**, the model performance significantly improves when the training sample size reaches 32 million. **Finally**, when the training sample size reaches 60 million, the model begins to converge, and the performance does not differ much until 131 million.
>
>
> **Q4: Further clarity on technical details of Minimum Description Length is needed.**
>
> > Weakness 3: The technical section should be clear and self-contained. Although the concept of Minimum Description Length (MDL) originates from existing studies, further clarity is needed on its technical details, such as how it is computed in the proposed method. At present, the technical content is somewhat difficult to follow.
>
> **Response:** Although minimum description length (MDL) is an abstract and incomputable concept in many fields, in symbolic regression it is quite simple: for a formula $f$, its length (i.e., the total number of mathematical symbols in it) is the MDL of the data $(x, y)$ corresponding to this formula. For example, $f(x) = x^2 + \sin(x)$ contains five symbols: $[x, \square^2, +, \sin, x]$, so the MDL of data $(x, y=f(x))$ is $5$. Therefore, although it is difficult to directly calculate the MDL from input data (that's why we introduce MDLformer), it is quite simple to calculate MDL from data generated by known formulas.
>
> We have revised **Section 3.2** and **Appendix B** of our article to add further technical details of MDL.
>
>
> **Q5: The differences and advantages of the proposed method to recent studies that adopt neural networks for symbolic reasoning should be discussed.**
>
> > Question 2: Some related studies should be discussed. Many recent studies [1] focus on adopting neural networks for symbolic reasoning. The authors should provide a brief discussion about the differences and advantages of the proposed method.
> > [1] A-NeSI: A Scalable Approximate Method for Probabilistic Neurosymbolic Inference. NeurIPS 2023.
>
> **Response:** We have read the paper [1] (https://arxiv.org/abs/2212.12393) you mentioned. It lies in the field of neural symbolic reasoning, which, despite the similar names, is a different task than symbolic regression.
>
> Specifically, this field focuses on the problems that, between the input feature $\mathbf{x}$ and target value $\mathbf{y}$, there exists a symbolic "concept" $\mathbf{w}$ that determines $\mathbf{y}$ in a known way $\mathbf{y}=c(\mathbf{w})$. The knowledge of $c$ can thus be used to help predict $\mathbf y$ and reason $\mathbf w$ behind the input $\mathbf{x}$. In the symbolic regression task, however, no such symbolic concept exists between the input data $(x,y)$ and the target formula $f$. This makes [1] and other neural symbolic reasoning methods unsuitable for symbolic regression tasks.
>
> We have revised our article to add the discussion about the differences between [1] and our method in Section 2 and Appendix A.

---

> ### Author Response · Authors · 2024-11-22
> **Willing to further clarify your remaining concerns**
>
> We appreciate your invaluable time and insightful comments. We have provided more details to your questions and concerns and added experiments on your provided benchmark as well as model performance under different amounts of training data. Can you kindly check them and let us know if they address your concerns? If you have further comments, we are happy to have a discussion. Thank you very much!

---

> > ### Comment · Reviewer_s6CS · 2024-11-24
> >
> > Thank you for your response. I have no further questions. Considering the authors' clarification, I have updated the score to 6.

---

### Official Review · Reviewer_1vTX · 2024-10-31

**Soundness:** 3
**Presentation:** 3
**Contribution:** 4
**Rating:** 8
**Confidence:** 4

**Summary:**

Existing symbolic regression frameworks operate by searching through the space of functions as vectors, minimizing a norm on the input-output behavior of the function. This paper instead approximates a search on the syntactic space of functions by computing a proxy metric for the minimum description length of the function necessary to take a given set of subcomponents and compute the full function. To do this, a synthetic dataset of functions is created and used to train a model that predicts a matching between functions and I/O behavior. The same dataset is then used to finetune this model to predict minimum description length. Overall, the model generally outperforms existing baselines, with particularly good relative performance in the case of noise.

**Strengths:**

This paper has a very elegant approach to the problem of symbolic regression, taking a principled approach towards the search process. Despite the theoretically intractable nature of the objective function selected, in practice it seems that a transformer is able to learn it well enough to make this search strategy better than existing strategies relying on end-to-end behavior. The resulting framework is more robust to noise than existing frameworks and more generalizable to a variety of domains. The results seem quite strong, with the only unfavorable result being underperformance relative to AIFeynman2 on the Feynman dataset in the absence of noise, which seems like a fairly narrow and understandable failure case.

**Weaknesses:**

Firstly, the datasets being evaluated on seem relatively small and thus potentially prone to being covered by the dataset. However, this seems to be a relatively standard convention that the related work follows, so I am not particularly concerned about this.

Secondly, the example in Figure 6 seems to suggest that there are very few alternate formulas that could be selected, and yet the recovery rate of this algorithm on the Feynman dataset is below 50%. Is this example atypical or is the search tree large even when heavily pruned? An explanation either way would be helpful (e.g., raw fractions for what the 4%, 0.2%, etc., are might clarify this if e.g., 4% is still 100 possibilities).

MInor:

Figure 6 and description: the blue sector description should probably say “all T that are lower according to the model than T^*” and emphasize that these are incorrect search paths -- it is a bit confusing to read otherwise. The caption does this better than the label on the  graph itself or the paragraph describing the figure.
471-472: “finding that nearly few” I think you mean “very few” or “nearly none”.
482: “for each of the three wheres” I think you mean “for each of the three strategies”
Figure 7: there are no axis labels for the right axis for d e and f.

Extremely minor nit: for figure 7 when you have two axes you color the left axis labels black and the right axis green, while the lines are blue and green. The axis labels should either both be black or blue and green.

**Questions:**

You say you sample D in line 215, which makes sense, since you need D to create the formula. However, you again sample D in line 221. However, here, shouldn’t D be the dimensionality of f?

I assume that the auxiliary loss on line 252 is computed per-batch? If so, this should be noted. If not, I would like more details on how this is done.

---

> ### Author Response · Authors · 2024-11-19
> **Response to Reviewer 1vTX**
>
> Thanks for your feedback. We have summarized your comments into four questions. We hope the following responses can address your concerns.
>
>
> **Q1: The evaluation datasets seem relatively small.**
>
> > Weakness 1. Firstly, the datasets being evaluated on seem relatively small and thus potentially prone to being covered by the dataset. However, this seems to be a relatively standard convention that the related work follows, so I am not particularly concerned about this.
>
> **Response:** **First**, symbolic regression traditionally focused on the capability of finding formulas on specific problems that people are interested in, rather than on the ability to consistently find formulas on arbitrary corner-case, so benchmarks in SR usually contain tens to a few hundred of questions, like SRBench (255 problems, 2022), Livermore (22 problems, 2021), and Nguyen (12 problems, 2011). **Secondly**, the validation dataset we used is actually the largest among related work:
> * Ours: **255** problems (119 Feynman, 14 Strogatz, 122 Black-box)
> * RSRM *(Xu, ICLR’24)*: **58** problems
> * SNIP *(Meidani, ICLR’24)*: **190** problems
> * SPL *(Sun, ICLR’23)*: **30** problems
> * E2ESR *(Kamienny, NeurIPS’22)*: **190** problems (and 100k problems generated in the same way as the training data, which aims to test the in-domain performance, so we did not count it. The same goes for DGSR and NeurSR.)
> * DGSR *(Kamienny, ICML'23)*: **190** problems (and 1k generated problems)
> * NeurSR *(Biggio, PMLR’21)*: **64** problems (and 200 generated problems)
> * DSR *(Petersen, ICLR'21)*: **37** problems
>
>
> **Q2: Is the example in Figure 6 atypical or is the search tree large even when heavily pruned?**
>
> > Weakness 2. Secondly, the example in Figure 6 seems to suggest that there are very few alternate formulas that could be selected, and yet the recovery rate of this algorithm on the Feynman dataset is below 50%. Is this example atypical or is the search tree large even when heavily pruned? An explanation either way would be helpful (e.g., raw fractions for what the 4%, 0.2%, etc., are might clarify this if e.g., 4% is still 100 possibilities).
>
> **Response:** The formula involved in Figure 6 is indeed an atypical example that only contains arithmetic operators. The raw fractions of 4% and 0.2% are 26/647 and 2/943 respectively, which is not large since there are only a few dozen incorrect search paths. But for other complex problems in the Feynman dataset, the trigonometric, exponential, or logarithmic operators contained in the target formulas can significantly affect the sample distribution and thus reduce MDLformer's prediction performance. For example, if $d_0 \sim \mathcal{N}(0,1)$, the distribution of $d_1=\exp(d_0)$ can deviate from $\mathcal{N}(0,1)$ greatly, reducing the model's performance in predicting the MDL for $(T(d_1), y)$.
> <!-- In this case, the model's prediction for $(T(d_1),y)$ will deteriorate. -->
> <!-- For example, for the `feynman_test_1` problem, there can be 29%=652/2240 and 37% = 628/1703 incorrect search paths in the first two steps. -->
>
>
> **Q3: You sample D in line 221 again. Shouldn’t D be the dimensionality of f sampled in line 215?**
>
> > Question 1. You say you sample D in line 215, which makes sense, since you need D to create the formula. However, you again sample D in line 221. However, here, shouldn’t D be the dimensionality of f?
>
> **Response:** You are right, the $D$ in line 221 is exactly the $D$ sampled in line 215. We have revised our article accordingly to avoid misunderstandings.
>
>
> **Q4: I assume that the auxiliary loss on line 252 is computed per batch.**
>
> > Question 2. I assume that the auxiliary loss on line 252 is computed per-batch? If so, this should be noted. If not, I would like more details on how this is done.
>
> **Response:** Yes, the auxiliary loss on line 252 is computed per batch. We have revised our article to note it.
>
>
> Once again, thank you for your constructive review and recognition of our work. We have revised the label and caption of Figure 6, as well as the corresponding description in the paragraph (Minor 1) to describe it more clearly. We also fixed the grammar issues (Minor 2,3), as well as the missing axis label (Minor 4) and unmatched axis color (ExMinor) in Figure 7d-f. The revised PDF has been uploaded.

---

> > ### Comment · Reviewer_1vTX · 2024-11-19
> >
> > Thank you for your response. I think the point about similar papers having small evaluation sets is reasonable, though I will see what other reviewers have to say. However, I do think that Figure 6 should be changed to depict a more typical example, or it being atypical should be mentioned, with explanation, in either the text or caption.

---

> > > ### Author Response · Authors · 2024-11-21
> > >
> > > Thanks for your suggestion. We have changed Figure 6 to a more typical example and provided more case studies in Appendix E.2.4 about problems with successes and failures recovery and problems from different datasets.

---

### Official Review · Reviewer_LjPc · 2024-11-04

**Soundness:** 2
**Presentation:** 3
**Contribution:** 2
**Rating:** 3
**Confidence:** 4

**Summary:**

This paper proposes a search objective based on the principle of minimum description length, which aims to streamline and enhance symbolic regression tasks. The authors generate a targeted training dataset and employ neural network techniques to develop a symbolic regression function. By using neural networks in this way, the approach is intended to improve both the efficiency and effectiveness of modeling complex data relationships through symbolic regression.

**Strengths:**

1. This paper is well-written, with the authors presenting the algorithm process and their perspectives clearly and comprehensively.
2. Figures 1 and 2 effectively illustrate the authors' viewpoints in a clear and intuitive way.
3. The proposed method demonstrates efficiency and achieves state-of-the-art performance compared to baseline methods.

**Weaknesses:**

1. This paper demonstrates limited innovation. The proposed method comprises three main parts: the first part provides a conventional description of neural network structure; the second generates a dataset, but the symbolic formula generation method follows that of Kamienny et al. (2022). The third part introduces two loss functions in the training process, with the first being the standard mean square error. Overall, the paper offers little originality.
2. Could larger datasets be used for experimentation? The datasets in this study are relatively small, such as Strogatz (14 problems) and Feynman (119 problems).
3. The proposed method relies on training with a substantial dataset (1.3 billion), while other methods have not been trained on datasets of this scale. This raises the question of whether the method’s effectiveness is largely attributable to the extensive volume of training data.

**Questions:**

1. In Table 1, the second-best methods are DSR (Petersen et al., 2021) and AIFeynman2 (Udrescu et al., 2020). These methods from 2020 and 2021 not only outperform those from 2022, 2023, and 2024 significantly but also achieve results up to four times greater. Could this discrepancy be due to variations in experimental settings?
2. The proposed method is trained on an extensive dataset (1.3B), while the other methods were trained on considerably smaller datasets. Could an ablation study be conducted, where all methods are trained on the same dataset to enable a more direct performance comparison?

---

> ### Author Response · Authors · 2024-11-19
> **Response to Reviewer LjPc (part 1)**
>
> Thanks for your feedback. We have summarized your comments into four questions. We hope the following responses can address your concerns.
>
> **Q1: What is the innovation and contribution of this work?**
>
> > Weakness 1. This paper demonstrates limited innovation. The proposed method comprises three main parts: the first part provides a conventional description of neural network structure; the second generates a dataset, but the symbolic formula generation method follows that of Kamienny et al. (2022). The third part introduces two loss functions in the training process, with the first being the standard mean square error. Overall, the paper offers little originality.
>
> **Response:** In this work, we propose an elegant approach for symbolic regression that significantly improves the success rate of recovering the accurate form of the target formula. The key idea is to search for a function $f$ acting on the $x$ to minimize the minimum description length (MDL) between $f(x)$ and $y$, rather than the accuracy used in existing work. The validity of this idea comes from the fact that, if the target formula $f^*$ satisfies that $y=f^*(x)=\phi(f(x))$, then the MDL between $f(x)$ and $y$ represents the length of $\phi$, and thus measures the "distance" between $f$ and the target. For example, if the target formula is $y=\sin(x^2+x)$ and we have searched for a formula $f(x)=x^2+x$, then the estimated MDL is 1 (since $\phi=\sin(\square)$ only contains one symbol "sin"), suggesting $f$ is very close to the target formula; with the error-based guidance, however, there is a huge difference between $f$ and the target.
>
> All of our designs serve this core idea. To estimate the MDL of any given data $(x,y)$, we use a transformer-based architecture, capable of predicting symbolic properties, as demonstrated in previous works. To train MDLformer, we generate synthetic formulas and corresponding data in a way that guarantees the diversity of generated data, and use the MSE between the predicted MDL and the ground truth as the loss function. To further improve the prediction performance of MDLformer, we use a loss function that aligns the numeric embedding space with the symbolic embedding space.
>
> Therefore, the three parts you mentioned are not merely pieced together but are integrated to collectively serve an innovative core concept: MDL-guided search.
>
>
> **Q2: Could larger datasets be used for evaluation experiments?**
>
> > Weakness 2. Could larger datasets be used for experimentation? The datasets in this study are relatively small, such as Strogatz (14 problems) and Feynman (119 problems).
>
> **Response:** **First**, in addition to the 133 ground-truth problems (14 Strogatz and 119 Feynman) you mentioned, we also evaluate our method on 122 black-box problems in Section 5.2. These problems are collected from the real world, rather than generated by any known formula, so they can measure SR methods' generalization capability. **Secondly**, just as Reviewer 1vTX said, this is a standard convention that the related work follows. Here we list the size of the validation dataset used by related work in recent years, where you can see that our method uses the largest evaluation dataset.
> * Ours: **255** problems
> * RSRM *(Xu, ICLR’24)*: **58** problems
> * SNIP *(Meidani, ICLR’24)*: **190** problems
> * SPL *(Sun, ICLR’23)*: **30** problems
> * E2ESR *(Kamienny, NeurIPS’22)*: **190** problems (and 100k problems generated in the same way as the training data, which aims to test the in-domain performance, so we did not count it)
> * DGSR *(Kamienny, ICML'23)*: **190** problems (and 1k generated problems)
> * NeurSR *(Biggio, PMLR’21)*: **64** problems (and 200 generated problems)
> * DSR *(Petersen, ICLR'21)*: **37** problems

---

> ### Author Response · Authors · 2024-11-19
> **Response to Reviewer LjPc (part2)**
>
> **Q3: Is it fair to use a large amount of training data? Since other methods are trained on considerably smaller datasets.** (Weakness 3 & Question 2)
>
> > Weakness 3. The proposed method relies on training with a substantial dataset (1.3 billion), while other methods have not been trained on datasets of this scale. This raises the question of whether the method’s effectiveness is largely attributable to the extensive volume of training data.
> > Question 2. The proposed method is trained on an extensive dataset (1.3B), while the other methods were trained on considerably smaller datasets. Could an ablation study be conducted, where all methods are trained on the same dataset to enable a more direct performance comparison?
>
> **Response:** **First**, we train our model with **131 million** = 0.131 billion data, rather than 1.3 billion. **Secondly**, not all symbolic regression methods are trained on considerably small datasets. An important class of symbolic regression methods -- neural network-based methods -- are trained on very large datasets, whose training dataset sizes are not much different in magnitude from ours:
> * NeurSR *(Biggio, PMLR’21)*: **100 million**
> * E2ESR *(Kamienny, NeurIPS’22)*: **38.4 million**
> * SNIP *(Meidani, ICLR’24)*: **56.3 million**
>
> **Thirdly**, different from existing neural network-based methods that learn to generate symbolic tokens in formulas, our method learns to predict lengths of formulas and thus does not have the contamination problem. In our symbolic regression method, the trained MDLformer can only estimate the formula length to provide the search objective, but cannot cheat by memorizing and reproducing the formulas in the training dataset -- since it has never seen the formulas in the training dataset but only their lengths.
>
> **Finally**, we conduct an ablation study in Figure 11 (**Appendix E.2.3**). When MDLformer is trained with the same training dataset size as in E2ESR, SNIP, and NeurSR, our model has a recovery rate of 42%, 66%, and 67% (on noise-free strogatz), still much higher than these methods (recovery rate <10%).
>
>
> **Q4: Why methods from 2020 and 2021 can outperform those from 2022, 2023, and 2024 significantly?**
>
> > Question 1. In Table 1, the second-best methods are DSR (Petersen et al., 2021) and AIFeynman2 (Udrescu et al., 2020). These methods from 2020 and 2021 not only outperform those from 2022, 2023, and 2024 significantly but also achieve results up to four times greater. Could this discrepancy be due to variations in experimental settings?
>
> **Response:** We used the official implementation of E2ESR(2022), SNIP (2023), and RSRM (2024), as well as the model weights they provided. The other running conditions are the same as the original articles, so there should be no issues about variation in experimental settings. **E2ESR (2022)** and **SNIP (2023)** are designed to generate formulas with high fitting accuracy, which is a different metric to the recovery rate that we used, and their generative paradigm makes them unable to find the correct formula form by running for a longer time, unlike DSR and AI-Feynman2 (as well as ours) which use search paradigm. **RSRM (2024)** was not evaluated on the SRBench in its original article, which may be because it is designed to discover formulas from sparse data (20~100 sample points, according to the original article). On the other hand, the Feynman and Strogatz datasets we used have more sample points and more complex formula forms, which may explain its performance issues.
>
> We have revised the experimental results part (**Section 5.1**) to reflect the discussion above.

---

> ### Author Response · Authors · 2024-11-22
> **Willing to further clarify your remaining concerns**
>
> We appreciate your invaluable time and insightful comments. We have provided more details to your questions and concerns and added experiments on different amounts of training data. Can you kindly check them and let us know if they address your concerns? If you have further comments, we are happy to have a discussion. Thank you very much!

---

> ### Author Response · Authors · 2024-11-28
>
> Dear Reviewer LjPc,
>
> Thank you for your thoughtful feedback and valuable suggestions, which have significantly helped us improve our work. We have carefully addressed your four questions regarding innovation and contribution, evaluation dataset size, training data amount, and baseline performance. Additionally, based on insightful comments from other reviewers, we have further enhanced the paper by adding supplemental experiments, including an analysis of model performance under different training data amounts and ablation studies on our data generation method.
>
> To ensure our contributions are clearly articulated, we have also added the **general comments** at the top of the page, which provides a more detailed discussion of the training data amount and highlights the key contributions of our work to symbolic regression algorithms, which have been positively received by other reviewers.
>
> **We kindly invite you to review these updates at your convenience. Please let us know if you have any further questions or suggestions—we would be delighted to address them.**

---

> > ### Comment · Reviewer_LjPc · 2024-12-01
> >
> > Dear Authors,
> >
> > I still have several additional questions regarding your submission after checking your response and revised draft:
> >
> > Firstly, the description of the innovation in your work could be clearer. It seems that the main contribution is the training of a larger, higher-quality dataset and the use of a combination of two loss functions to train a neural network. The concept of training a neural network with multiple loss functions is well-recognized in the deep learning community. Additionally, it is generally anticipated that using a larger, higher-quality dataset will result in improved experimental outcomes. The results in Fig. 11 also indicate that the performance of MDLformer degrades notably with a reduction in the number of training samples.
> >
> > Secondly, simply equalizing the number of training samples across datasets is not adequate, as the training datasets vary not only in size but also in quality. Training on higher-quality datasets will naturally yield better results. If the primary contribution of the authors is the generation of a dataset that is both larger and of higher quality, the concern about differing quality is understandable.
> >
> > Thirdly, the authors stated in their response: "We used the official implementation of E2ESR (2022), SNIP (2023), and RSRM (2024), as well as the model weights they provided. The other running conditions are the same as the original articles." If the settings and model weights are identical to those in the original papers, then the comparison methods are based on their original datasets. As previously mentioned, different training datasets have varying quantities and qualities. It is currently challenging to ascertain whether the observed experimental improvements are due to the dataset or the MDLformer method itself. To accurately compare the performance of different methods, it is crucial that they are trained on the same dataset, preferably not one generated by the authors. This is because the authors' dataset is tailored to minimize description length, whereas other datasets are not designed with this objective in mind.
> >
> > Fourthly, the authors mentioned in their response: "E2ESR (2022) and SNIP (2023) are designed to generate formulas with high fitting accuracy, which is a different metric from the recovery rate that we used." Apart from these two methods, we note that other studies use the R2-score, which seems to differ from the metric used in your paper, as there are inconsistencies in the reported values for the same dataset. It appears that the comparison is made using an unconventional metric, which is somewhat unusual. Shouldn't a more widely accepted metric be used, especially given that many other methods adopt the R2-score as a standard metric?

---

> > > ### Author Response · Authors · 2024-12-01
> > > **Further Response to Reviewer LjPc (part 3)**
> > >
> > > **Q3: Shouldn't the more widely accepted R2 be used as the metric?**
> > >
> > > > Fourthly, the authors mentioned in their response: "E2ESR (2022) and SNIP (2023) are designed to generate formulas with high fitting accuracy, which is a different metric from the recovery rate that we used." Apart from these two methods, we note that other studies use the R2-score, which seems to differ from the metric used in your paper, as there are inconsistencies in the reported values for the same dataset. It appears that the comparison is made using an unconventional metric, which is somewhat unusual. Shouldn't a more widely accepted metric be used, especially given that many other methods adopt the R2-score as a standard metric?
> > >
> > > Thank you for raising this point. **First, the recovery rate is a suitable and conventional metric**. In this work, we point out that in evaluation problems with ground truth formulas, a symbolic regression algorithm should be capable of discovering formulas with the correct mathematical form, rather than merely settling for finding sufficiently accurate formulas. Therefore, we use the recovery rate as the metric. Furthermore, the recovery rate is not an unconventional metric, but is widely used by many works, to name a few:
> > > * Reinforcement Symbolic Regression Machine (**[ICLR'23](https://arxiv.org/pdf/2305.14656#page=7)**),
> > > * Deep Generative Symbolic Regression (**[ICLR'23](https://arxiv.org/pdf/2401.00282#page=9)**)
> > > * Deep symbolic regression: Recovering mathematical expressions from data via risk-seeking policy gradients (**[ICLR'21](https://arxiv.org/pdf/1912.04871#page=8)**)
> > > * Contemporary symbolic regression methods and their relative performance (i.e. SRBench, **[NeurIPS'21](https://arxiv.org/pdf/2107.14351#page=9)**)
> > >
> > > **Secondly, we have provided the $R^2$ results in Appendix [Tables 4~7](https://openreview.net/pdf?id=ljAS7cPAU0#page=19), along with the Pareto front between $R^2$ and formula length in Appendix [Figure 8](https://openreview.net/pdf?id=ljAS7cPAU0#page=18)**. We apologize for not explicitly mentioning this in the main text. From Tables 4~7, it can be seen that our method achieved the highest $R^2$ at two noise levels and the second-highest $R^2$ at one noise level in the Strogatz benchmark with four different noise levels. On the Feynman problem set, although our method does not achieve the highest $R^2$ ranking, it discovers formulas with simpler forms compared to methods with higher $R^2$ rankings. Therefore, as shown in Figure 8, our method is positioned in the first rank of the Pareto front that balances formula accuracy and complexity.
> > >
> > > **Thirdly**, for black-box problems where the correct formula is not known, $R^2$ is an important metric. Therefore, **we used the $R^2$ as the metric for black-box problems in [Figure 5](https://openreview.net/pdf?id=ljAS7cPAU0#page=8)**. The results show that our method is positioned in the first rank of the Pareto front that balances formula accuracy and complexity.
> > >
> > > ---
> > >
> > > We hope this clarification can address your questions. Please let us know if you have any further questions or need additional clarification.

---

> ### Author Response · Authors · 2024-12-01
> **Further Response to Reviewer LjPc (part 1)**
>
> Thank you for your valuable feedback. We have carefully reviewed your comments and have provided the following clarifications in response to your questions. We hope these answers address your concerns effectively.
>
> **Q1: What is the contribution of this work?**
>
> > Firstly, the description of the innovation in your work could be clearer. It seems that the main contribution is the training of a larger, higher-quality dataset and the use of a combination of two loss functions to train a neural network. The concept of training a neural network with multiple loss functions is well-recognized in the deep learning community. Additionally, it is generally anticipated that using a larger, higher-quality dataset will result in improved experimental outcomes.
>
> **First**, the innovation of our work is not “training of a larger, higher-quality dataset and the use of a combination of two loss functions to train a neural network”. Instead, our contribution is **a new symbolic regression framework** that uses the minimum description length (MDL) as the search objective. This framework contains two parts:
> 1) an **MDLformer** that is trained on a synthetic dataset to estimate the MDL of given x-y pairs, and
> 2) an **MDL-guided symbolic regression algorithm** that finds the correct formula describing the input x-y pairs using the MDLformer's estimation as the search objective.
>
> **Secondly**, to further illustrate our innovation, we compare the proposed method with two types of existing methods: 1) heuristic search methods and 2) neural network-based generative methods, as shown in the **[Online Figure](https://anonymous.4open.science/r/SR4MDL-5CF3/comparasion.pdf)**. As illustrated in the figure, our method differs from heuristic search methods in terms of its search objective, which shifts from accuracy to the MDL estimated by MDLformer. On the other hand, unlike neural network-based generative methods, where the trained transformer directly generates the symbolic sequence in an end-to-end manner, our proposed method incorporates the MDLformer as a component of the framework, which provides estimated MDL values that serve as search objectives for the search algorithm.

---

> ### Author Response · Authors · 2024-12-01
> **Further Response to Reviewer LjPc (part 2)**
>
> **Q2: Unfair performance comparison with baseline methods in terms of training dataset.**
>
> > Secondly, simply equalizing the number of training samples across datasets is not adequate, as the training datasets vary not only in size but also in quality. Training on higher-quality datasets will naturally yield better results. If the primary contribution of the authors is the generation of a dataset that is both larger and of higher quality, the concern about differing quality is understandable.
> >
> > Thirdly, the authors stated in their response: "We used the official implementation of E2ESR (2022), SNIP (2023), and RSRM (2024), as well as the model weights they provided. The other running conditions are the same as the original articles." If the settings and model weights are identical to those in the original papers, then the comparison methods are based on their original datasets. As previously mentioned, different training datasets have varying quantities and qualities. It is currently challenging to ascertain whether the observed experimental improvements are due to the dataset or the MDLformer method itself. To accurately compare the performance of different methods, it is crucial that they are trained on the same dataset, preferably not one generated by the authors. This is because the authors' dataset is tailored to minimize description length, whereas other datasets are not designed with this objective in mind.
> >
> > The results in Fig. 11 also indicate that the performance of MDLformer degrades notably with a reduction in the number of training samples.
>
> Thank you for your feedback. In fact, our method does not use a higher-quality dataset compared to existing methods. And we have conducted an ablation experiment for the training data amount in the modified article based on your suggestion to make a fairer comparison.
>
> ***1) Same data generation method and data quality***. Our work does not feature a higher-quality dataset. In fact, **we generate the training data in the same way as other neural network-based generative methods** (E2ESR and SNIP). Therefore, the training data does not differ from these baseline methods in terms of quality. The only difference lies in the labels used in the training process: with generated formula $f$ and x-y pairs, existing generative methods predict symbols in the formula based on input x-y pairs, while our MDLformer predicts the length of the formula based on input x-y pairs. For example, consider the generated training data $f(x) = x+1, \mathbf x = [1, 2, 3, 4]^T, \mathbf y = [2, 3, 4, 5]^T$. This data is generated in the same manner for both our method and the baseline methods. However, the baseline methods train a transformer $M$ to predict
> $$
> \begin{aligned}
> &M(\mathbf x, \mathbf y, []) = x, \\\\
> &M(\mathbf x, \mathbf y, [x]) = +, \\\\
> &M(\mathbf x, \mathbf y, [x, +]) = 1, \\\\
> &M(\mathbf x, \mathbf y, [x, +, 1]) = <\text{End of Sequence}>,
> \end{aligned}
> $$
> while our MDLformer is trained to predict
> $$
> \text{MDLformer}(\mathbf x, \mathbf y) = 3,
> $$
> where $3$ corresponds to the number of symbols in the formula $f(x)$ (i.e., $x$, $+$, and $1$). **This difference is not related to the data quality but rather pertains only to the label formula used in training. Therefore, it does not result in a higher-quality dataset.**
>
> ***2）Different data amount***. To address the potential concern regarding the data amount, we conducted an ablation experiment, the results of which are presented in **[Figure 11](https://openreview.net/pdf?id=ljAS7cPAU0#page=26)**, from which we observe that:
>
> 1. **Performance gain originated from search objective shift**: When the data amount matches that used in E2ESR, SNIP, and NeurSR, the recovery rates reach 42%, 66%, and 67%, respectively, which is substantially higher than those of the baseline methods (<10%). This demonstrates that **the advantage of our approach lies in shifting the search objective from accuracy to MDL instead of the training data volume**.
> 2. **Necessity of large-scale training corpus**: The model’s performance improves significantly when the data amount reaches 32 million. This suggests that **MDLformer requires sufficient data to estimate MDL accurately**, highlighting the importance of a large-scale training corpus.
> 3. **Further increasing data volume is not promising**: The model begins to converge at a data amount of 60 million, and the performance does not differ much until 131 million. This indicates that further improvements in future work may **require optimizing the data generation process rather than simply increasing the data volume**.

---

> ### Author Response · Authors · 2024-12-03
>
> Dear Reviewer LjPc,
>
> Thank you again for your valuable time and thoughtful feedback. We hope our responses have addressed your remaining questions and concerns. Since the remaining time for reviewers to reply is less than 8 hours, we kindly request your feedback to confirm if there are any existing issues. Should there be no additional concerns, we would appreciate it if you could consider revising your score.

---

> > ### Comment · Reviewer_LjPc · 2024-12-03
> >
> > Dear Author,
> > I have carefully reviewed your response and paper. However, I remain unconvinced by the use of different training datasets for the comparative experiments. In the deep learning and machine learning communities, it is a well-established practice to use the same training dataset when evaluating the performance of different methods to ensure fair comparisons.
> > I find it problematic that the results were achieved using a self-generated, larger training dataset, as this approach resembles a data augmentation technique rather than a direct comparison.
> > Additionally, the dataset generation method used appears to have been directly adapted from existing work, raising further concerns regarding the contribution of the methodology. The experimental results demonstrate that this training dataset has substantially improved the outcomes. However, as the dataset was constructed using another researcher’s method, this raises additional questions about the contribution of your work.
> > To clarify, my concerns do not stem from differences in dataset quality or size, nor do I argue for higher-quality datasets.
> > In summary, the paper makes limited contributions in terms of network structure or dataset generation methodology.

---

> > > ### Author Response · Authors · 2024-12-04
> > >
> > > Thank you for providing your detailed comment.
> > >
> > > However, we respectfully disagree with your judgment on the contribution of our paper, which we explicitly described in both the original paper (L18-20) and the previous response. **Our main contribution is the new framework of symbolic regression by searching for minimum description length (MDL) instead of minimum prediction error (MPE)**. As we have plotted in [Online Figure](https://anonymous.4open.science/r/SR4MDL-5CF3/comparasion.pdf), this idea is completely different from existing approaches, including both search-based methods and end-to-end generative methods. Our proposed framework can ensure a monotonic search process in principle due to its optimal substructure: the search direction of MDL reduction is always consistent with the direction leading to the target formula (L80-83, Figure 1). To achieve this objective, we design a transformer-based module that can effectively learn to estimate MDL from a self-generated training dataset. **We do not claim to contribute to a new data generation method or a new network architecture for SR**. Instead, we demonstrate that by incorporating off-the-shelf techniques, our new framework can significantly improve SR performance. According to your suggestion in the initial comment, we also add ablation studies to demonstrate that **this performance gain comes from a paradigm shift (from MPE to MDL) and not the larger training dataset (Appendix E.2.3, Figure 11)**.
> > >
> > > Regarding your comment that “it is a well-established practice to use the same training dataset when evaluating the performance of different methods to ensure fair comparisons”, we have to point out that **existing DL-based SR works do not follow the requirement of using the same training dataset**. To name a few, NeurSR(ICML'21), E2ESR(NeurIPS'22), and SNIP(ICLR'24) use the same data generation method but with different data volumes, i.e., 100M, 38.4M, and 56.3M, respectively. We believe this is because, unlike traditional supervised learning settings, DL-based SR works normally choose to pretrain models on a synthetic dataset generated by a certain data distribution and then test on other benchmark problems with a different distribution. Nonetheless, **we value your initial comment that it is more fair to control the impact of dataset volume when evaluating different models**. Our results indeed verify that the main performance gain comes from our novel design of the search process instead of a larger dataset (Appendix E.2.3, Figure 11). Our further analysis emphasizes the limitation of increasing dataset volume (General Reclaim-part 2), which opens future research directions for MDL-based SR works regarding improving training data quality.
> > >
> > > Finally, we find some contradictory points in your comments. In your initial comment (review), you criticize that we generated the dataset using the same method as Kamienny et al. (2022), but with a different dataset size. After we conduct a fair experiment by controlling it the same as baselines, you further comment that we use a better-quality dataset. This is puzzling to us, as using the same data generation method as Kamienny et al. (2022) means that we cannot generate a dataset of better quality. We sincerely hope you can kindly check both our revised paper and response and be more positive about our submissions.

---

### Author Response · Authors · 2024-11-27
**General Reclaim (part 1)**

We sincerely appreciate the insightful comments provided by all reviewers, which helped us improve the article. We have replied to all concerns and significantly improved the manuscript. Here we would like to highlight the key novelty of our work as well as discuss the influence of the training data amount that most of the reviewers are concerned about.

***1. Key novelty***

Our key novelty lies in the shift of search objective from accuracy to minimum description length (MDL) in symbolic regression.

**First**, MDL reflects the "distance" between the searched formula and the target formula and thus decreases monotonically as the search gets closer to the target, while accuracy cannot (see **[Figure 1](https://openreview.net/pdf?id=ljAS7cPAU0#page=2)**). This makes MDL a better search objective than the accuracy used by previous works.

> - Reviewer 1vTX praised this idea as "very elegant" ("*This paper has a **very elegant** approach to the problem of symbolic regression*.")
> - Reviewer s6CS also commented that "*The overall idea of this paper is **intuitive and reasonable**. The traditional search objective, i.e., prediction error, cannot effectively capture the distance between the target and current formulas*."

**Secondly**, to deal with the uncomputable nature of MDL, we train an MDLformer to learn it on a synthetic training dataset with randomly generated formulas and $(x,y)$ pairs. We design a two-stage training scheme that first aligns the embedding spaces of symbolic formulas and numeric x-y pairs to improve cross-modal prediction performance and then trains the MDLformer to estimate the MDL of x-y pairs. Our result demonstrates that MDLformer can estimate MDL of given x-y pairs accurately (**[Table 2](https://openreview.net/pdf?id=ljAS7cPAU0#page=9)**) and robustly (**[Figure 7](https://openreview.net/pdf?id=ljAS7cPAU0#page=10)**).

**Finally**, the trained MDLformer can be used for symbolic regression. During the symbolic searching, we can get close to the target formula by iteratively selecting transforms that reduce the estimated MDL of the formula (**[Figure 6](https://openreview.net/pdf?id=ljAS7cPAU0#page=9)**). This leads to quite competitive results, with significantly improved formula recovery rates across different datasets and noise levels.

> - Reviewer 1vTX: "*The resulting framework is **more robust** to noise than existing frameworks and **more generalizable** to a variety of domains.*"
> - Reviewer s6CS: "*The experimental results are **promising**. The proposed method significantly outperforms existing methods, demonstrating that a more sophisticated objective can help improve the recovery rate.*"

---

> ### Author Response · Authors · 2024-11-27
> **General Reclaim (part 2)**
>
> ***2. Influence of the training data amount***
>
> Two reviewers (LjPc, s6CS) expressed concerns about the large amounts of training data and possible data contamination issues, with reviewer s6CS having already accepted our response. Here we summarize our clarification on this issue as follows:
>
> * **Close to other neural network-based methods**: Our model is trained with **131 million** data, which is close in magnitude to existing works like NeurSR (**100 million**), E2ESR(**38.4 million**), and SNIP (**56.3 million**).
> * **No data contamination issue**: Different from these existing works that directly learn symbolic tokens in formulas and thus can have data contamination issues, our model learns the length of the formula (i.e., minimum description length). Therefore, it has never seen the formulas in the training dataset but only their lengths, making it unable to cheat by memorizing and reproducing the formulas in the training dataset.
> * **Ablation study about training data amount**: In [Figure.11](https://openreview.net/pdf?id=ljAS7cPAU0#page=26) (Appendix E.2.3) we conduct an ablation study to evaluate MDLformer’s performance under varying data amounts, revealing the following insights:
>     1. **Necessity of large-scale training corpus**: The model’s performance improves significantly when the data size reaches 32 million. This suggests that **MDLformer requires sufficient data to accurately estimate MDL**, highlighting the importance of large-scale training corpus.
>     2. **Performance gain originated from search objective shift**: When the data size matches that used in E2ESR, SNIP, and NeurSR, the recovery rates reach 42%, 66%, and 67%, respectively, which is substantially higher than those of the baseline methods (<10%). This demonstrates that **the advantage of our approach lies in shifting the search objective from accuracy to MDL instead of the training data volume**.
>     3. **Further increasing data volume is not promising**: The model begins to converge at a data size of 60 million, and the performance does not differ much until 131 million. This indicates that further improvements in future work may **require optimizing the data generation process rather than simply increasing the data volume**.
>
>
> ---
>
> Let me know if you'd like further refinements!
>
> The revised part is highlighted as colored text in the manuscript.

---

### Meta-Review · Area_Chair_eVGX · 2024-12-23

**Metareview:**

This paper proposes a novel method for symbolic regression, namely discovering the symbolic formula fitting data. Unlike existing methods where the difference between the symbolic function and the actual data is minimized during the search steps, this paper proposed to minimize the minimum description length, a more appropriate metric for this task. Calculating the minimum description length is nontrivial, and the authors trained a transformer to do this calculation. The experiments show promising results on a few standard evaluation benchmarks in this field.

While two reviewers recognized the importance and novelty of this work, reviewer LjPc questioned the contribution of this work and the fairness of comparison against prior work. The AC has also carefully read the paper and the reviews, and believes that the review from LjPc was indeed mostly due to misunderstandings of its setting. In this symbolic regression setting, the final goal is to recover the original symbolic expressions with high accuracy, and many baselines used self-generated training examples to fit their own pipeline, so the training examples are incomparable. A better metric the authors should actually compare is the amount of compute used for building this pipeline, compared to prior work. Despite some concerns on evaluation and some improvements on writing should be done to avoid confusion (like those from reviewer LjPc), the AC believes the methodology proposed in this work is insightful and may inspire future research in this field, so we recommend the acceptance of this paper.

**Additional Comments On Reviewer Discussion:**

Reviewer 1vTX and s6CS acknowledged the author's rebuttal and clearly indicated their concerns were addressed. Reviewer LjPc had multiple rounds of discussions with the authors. After carefully reviewing the conversations, the AC believes that the key concern stated by reviewer LjPc was indeed a misunderstanding of this problem setting. The final recommendation was made with these considerations.

---

### Decision · Program_Chairs · 2025-01-22

Accept (Poster)